# Blue and green food webs respond differently to elevation and land use

Hsi-Cheng Ho [1] ✉, Jakob Brodersen[2], Martin M. Gossner [3,4], Catherine H. Graham[3], Silvana Kaeser[1], Merin Reji Chacko [3,4], Ole Seehausen [2,5], Niklaus E. Zimmermann [3,4], Loïc Pellissier [3,4] & Florian Altermatt [1,6] ✉

While aquatic (blue) and terrestrial (green) food webs are parts of the same landscape, it remains unclear whether they respond similarly to shared environmental gradients. We use empirical community data from hundreds of sites across Switzerland and a synthesis of interaction information in the form of a metaweb to show that inferred blue and green food webs have different structural and ecological properties along elevation and among various land-use types. Specifically, in green food webs, their modular structure increases with elevation and the overlap of consumers' diet niche decreases, while the opposite pattern is observed in blue food webs. Such differences between blue and green food webs are particularly pronounced in farmland-dominated habitats, indicating that anthropogenic habitat modification modulates the climatic effects on food webs but differently in blue versus green systems. These findings indicate general structural differences between blue and green food webs and suggest their potential divergent future alterations through land-use or climatic changes.

Biological communities are not just arithmetic sums of species. Instead, species therein interact with each other to form ecological networks that underpin the structure of biodiversity and ecological functions[1,2]. Therefore, besides looking at species richness and composition of communities, we need to additionally understand how ecological networks respond to key environmental drivers to better inform biodiversity management in this era of global change[1,3–5]. In this regard, climate and land use are well-known drivers to shape species richness and composition[6–8], but how the ecological networks may respond to these drivers remains a relevant and contemporary topic to be explored.

Recent studies have started to investigate ecological networks' structural changes along environmental gradients, as well as their effects on species persistence or ecosystem functioning[9–11]. However, these investigations mostly targeted interactions between two functionally distinctive taxonomic groups, e.g., bipartite herbivore-plant or host-parasitoid networks[12–14], or were restricted to simplified experimental systems[15]. Such partiality is likely due to methodological constraints, as measuring interactions in a standardised and comparable way across a broad range of taxonomic (or functional) groups is difficult, not to mention further across environmental gradients, which has hitherto remained elusive. Consequently, although consumer-resource trophic interaction is arguably the most essential biological interaction[16,17] associating virtually every species in any biome, we have a rather limited understanding of the responses of real-world multi-trophic food webs to the changing environment (see[18] for rare

[1]Department of Aquatic Ecology, Eawag: Swiss Federal Institute of Aquatic Science and Technology, Überlandstrasse 133, CH-8600 Dübendorf, Switzerland. [2]Department Fish Ecology and Evolution, Eawag: Swiss Federal Institute of Aquatic Science and Technology, Seestrasse 79, CH-6047 Kastanienbaum, Switzerland. [3]WSL Swiss Federal Research Institute, Zürcherstrasse 111, CH-8903 Birmensdorf, Switzerland. [4]Department of Environmental Systems Science, Institute of Terrestrial Ecosystems, ETH Zürich, Universitätstrasse 16, CH-8092 Zürich, Switzerland. [5]Division Aquatic Ecology, Institute of Ecology and Evolution, University of Bern, Baltzerstrasse 6, CH-3012 Bern, Switzerland. [6]Department of Evolutionary Biology and Environmental Studies, University of Zürich, Winterthurerstrasse 190, CH-8057 Zürich, Switzerland. ✉e-mail: hsichengho@gmail.com; Florian.Altermatt@eawag.ch

examples). Moreover, it is also unclear whether food webs of different biomes tend to respond differently.

Aquatic (blue) and terrestrial (green) food webs have mostly been studied separately, but both are fundamental parts of the same landscape[19]. Blue food webs tend to accommodate long food chains[20,21] and have marked large-eat-small body-size relationships between consumers and resources[22,23], thus often exhibiting a nested structure[22]. In contrast, green food webs have comparatively short food chains[20,21] with less-prominent body-size relationships[24], and generally exhibit a modular (compartmentalised) structure[24,25]. Despite knowing these typical structural differences, the pronounced separation between aquatic and terrestrial ecological research has prevented systematic comparisons between the two types of food webs in their responses to environment drivers (see[26] for a global-scale example). However, such understanding is crucial, since blue and green communities can be regulated by the same environmental factors[27], and landscape conservation strategies for both systems at common places need to be well-aligned to avoid lopsided management[28]. This knowledge gap is therefore most relevant, and also suitable to be addressed, at a landscape scale. At such a spatial scale, blue and green communities respectively have a shared regional species pool, thereby having comparable species composition among local food webs. Meanwhile, they both span shared environmental gradients, such as elevation and various forms of land use, thereby providing comparable factor ranges between the biomes.

To study the spatial variation in multi-trophic food webs, the combination of a trophic metaweb and empirical species co-occurrence can provide an efficient tool[29]. A metaweb is a representation of the regional food web integrating the knowledge of trophic interactions among species that are present in the target region[10,30]. With the assumption that every consumer species has a fixed ability to feed on certain resources (i.e., a fixed potential diet set) across the focal spatial range, the established trophic relationships in the metaweb allow an inference of local food webs in combination with local community composition[31] (see illustration in Fig. 1). Such inferential food-web construction implicitly embraces the concept that interactions are driven by matching functional characteristics, and collapses potential intraspecific variations of these characteristics at species level[30,32]. Therefore, the inferred local food webs essentially reflect a set of potential trophic interactions within realistic boundaries where species "known interacting" and "co-occurred". By coherently applying the same metaweb throughout, this approach can guide an unbiased identification of structural differences between potential food webs that result from compositional differences between local communities, thereby allowing the investigation of how food webs may change along environmental gradients. The metaweb approach is especially adequate for comparing blue and green food webs, as the two are inherently composed of very different taxa, and thus must be standardised for an adequate inference across both systems, for example based on species co-occurrence data.

To fill the mentioned knowledge gaps, we address (i) how multi-trophic food webs are influenced by key environmental drivers, and (ii) if blue and green food webs respond differently to these drivers. We thus examine the properties of blue and green food webs inferred using the metaweb approach. Specifically, we focus on a selected set of holistic food-web metrics based on their ecological relevance and past usage in the literature (see *Methods*), and compare how blue and green food webs respectively change along elevation and across major land-use types (i.e., forests, scrubs, open spaces, farmlands and urban areas). The elevation is chosen as a synthetic predictor since it encapsulates the effects of temperature as pre-analysed with our data (see *Methods* and *Discussion*). In general, food-web patterns discovered along elevation can be interpreted as a spatial proxy of their responses to temperature and its change, in combination with potential influences of some non-climatic drivers. Likewise, patterns among land-use types can be seen as responses to habitat turnover

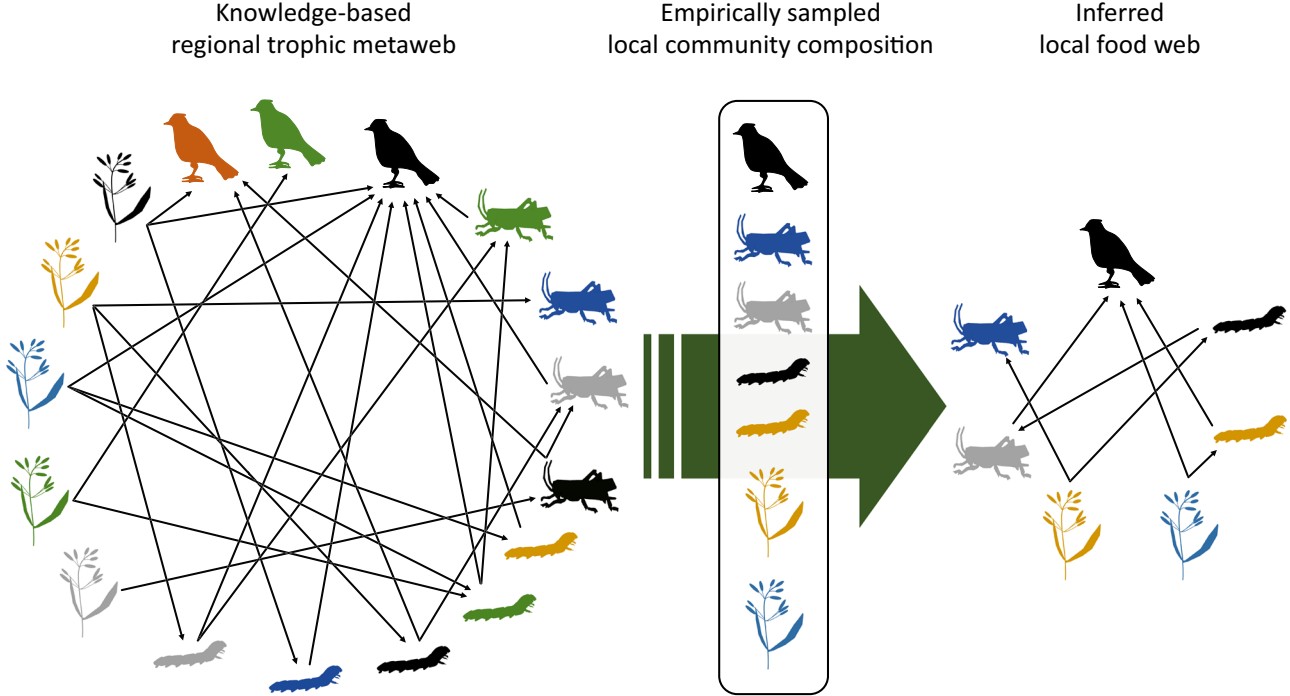

**Fig. 1 | An illustration of the metaweb approach using dummy terrestrial species.** Different silhouette shapes and colours represent different organismal groups and species, respectively, and black arrows indicate trophic relationships pointing from the resource to the consumer. Trophic relationships established in the metaweb are based on the literature and expert knowledge (left panel). Extracting trophic relationships among species that co-occur locally according to empirical survey data (middle panel) allows the inferential assembly of local food web (right panel).

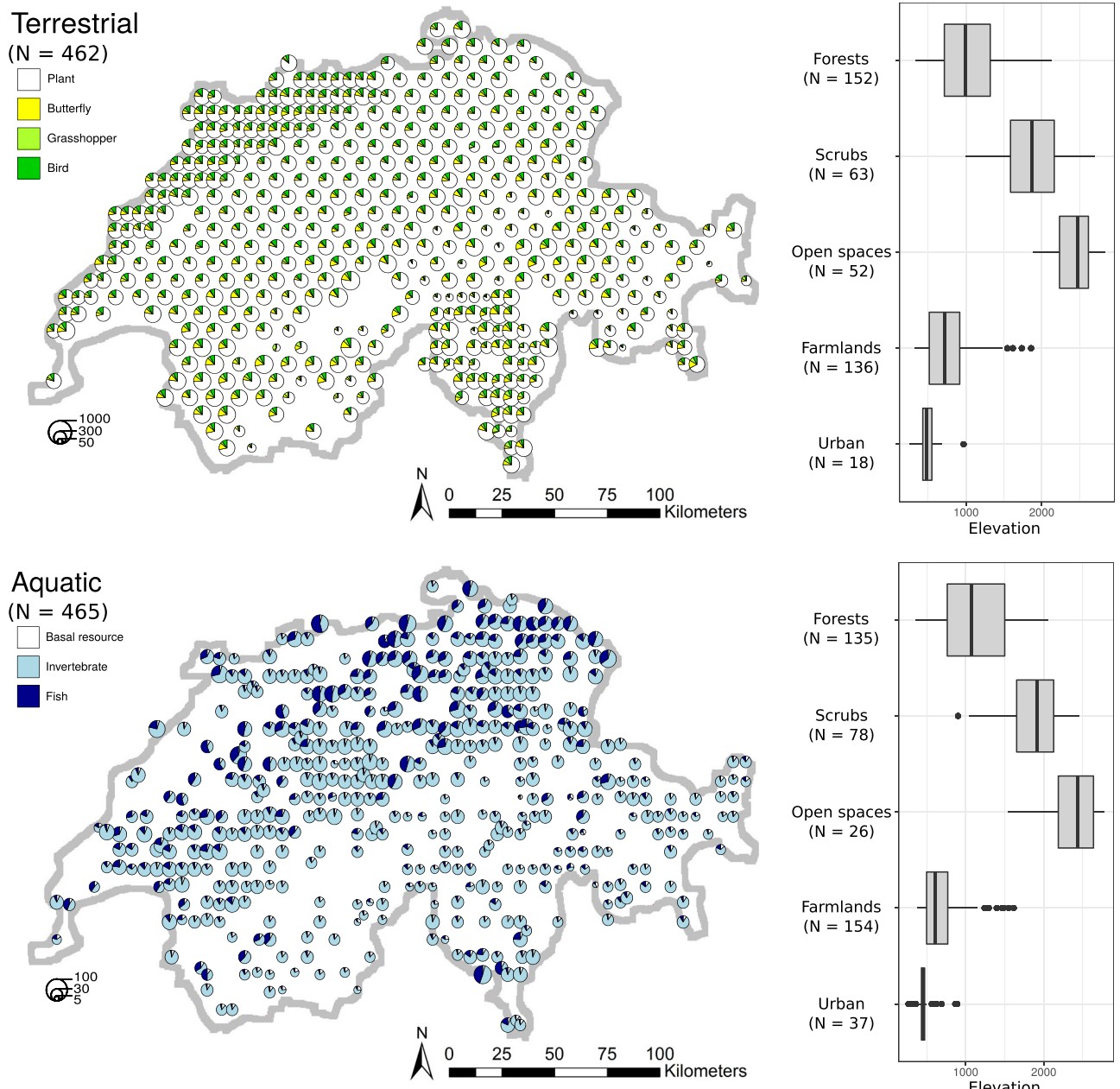

**Fig. 2 | Terrestrial (top panel) and aquatic (bottom panel) food webs assembled in this study.** The food webs' locations across Switzerland are based on a stratified, randomised raster approach, and thus representative for the landscape. The pie charts give the food webs' sizes (number of nodes, as the size of pie charts) and focal-group composition (colours). Note that the three assumptive mega nodes: plant, plankton, and detritus, served as basal resource in all aquatic food webs (*Methods*). The boxplots show, at where these food webs locate, how different dominant land-use types distribute along elevation. The boxes span lower to upper quartiles with middle lines denoting the medians, the whiskers indicate 1.5 inter-quartile range values, and the dots give respective outliers.

driven by climate and/or anthropogenic activities. To tackle these research questions at a landscape scale, we use empirical occurrence data from highly systematic and representative species-diversity monitoring schemes across freshwater and terrestrial biomes in Switzerland, which encompass the occurrence of aquatic inverte-brates and fishes, as well as terrestrial plants, butterflies, grass-hoppers, and birds over a 42,000 km² area. Our findings reveal divergent elevational patterns between the blue and green food webs, as well as their distinct land-use dependence in elevational responses. Blue and green food webs, therefore, respond differently to elevation and land use, where the two drivers actually have interactive impacts on food webs.

## Results
### Inference of food web
We inferentially assembled local food webs for a total of 462 terrestrial (green) and 465 aquatic (blue) sites, which representatively covered the area (42,000 km²) of Switzerland and spanned an elevational range from 249 m to 2834 m a.s.l. (Fig. 2). With increasing elevation, different land-use types dominated the vicinity of local sites, and their eleva-tional turnovers were consistently covered by the blue and green sites (Fig. 2). The inferred food webs included distinct taxa mostly at the species level of overall 2016 plant, 191 butterfly (focused on trophic interactions of larvae), 109 grasshopper, 155 bird, 248 stream inver-tebrate, and 78 stream fish taxa (henceforth "focal groups"; see

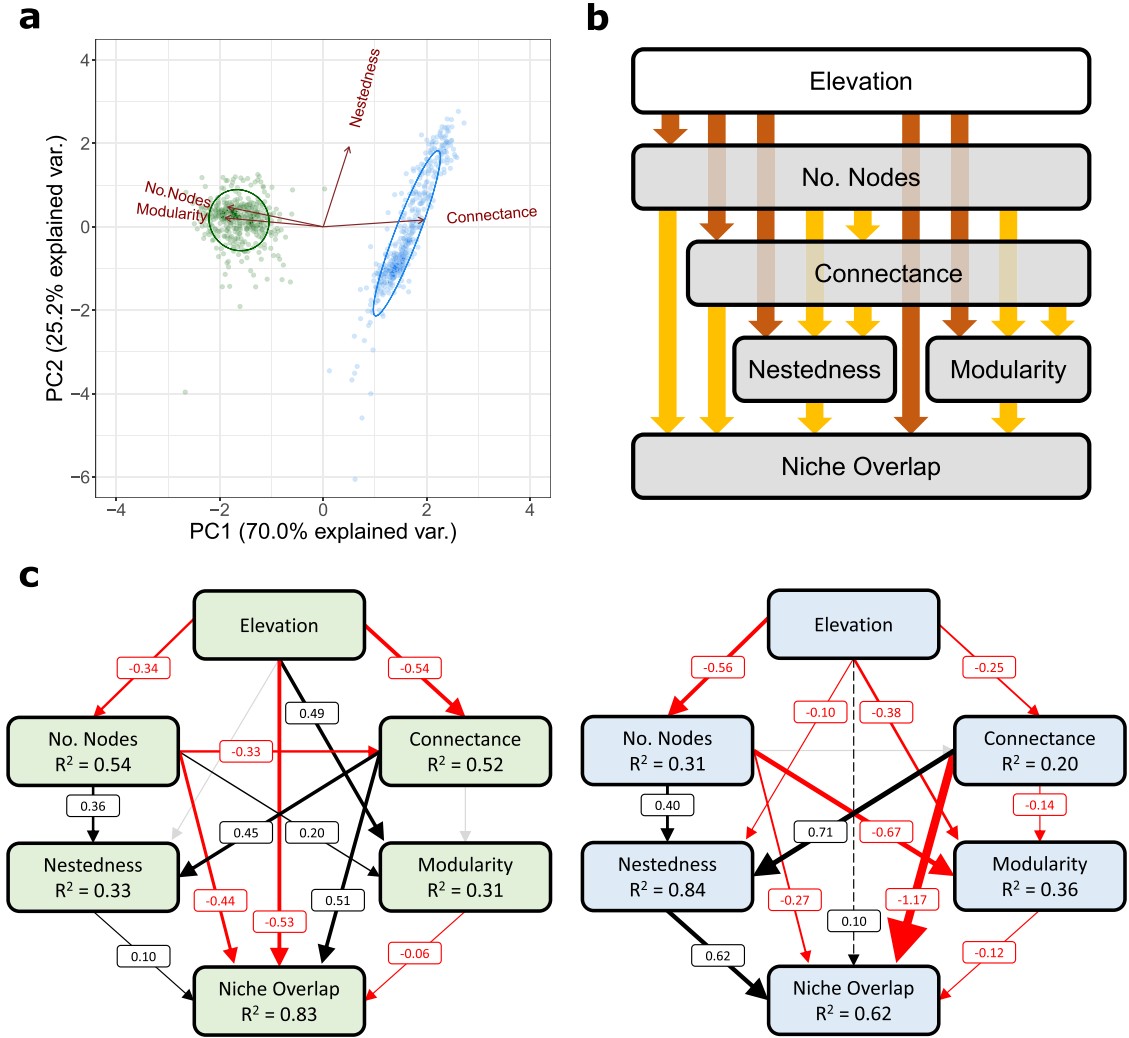

**Fig. 3 | Structural properties of terrestrial and aquatic food webs and the potential elevational influences on them. a** The principal component analysis reveals distinct structures between the terrestrial (green) and aquatic (blue) food webs, indicated by dot colours. **b** The potential dependencies between elevation and food-web properties (brown arrows), as well as among food-web properties themselves (yellow arrows), for the piecewise structural equation modelling (SEM) analysis. **c** The outcomes of piecewise SEM standardised coefficients and R-squared of the green and blue food webs (indicated by block colours), respectively. Positive paths in black, negative in red, marginal-significant as dashed, and non-significant in grey. A path's width is proportional to its size of standardised coefficient. For additional SEM results, see Supplementary Figs. 1, 2, Supplementary Tables 7–12.

*Methods* for details). The blue and green food webs differed in their structural properties (i.e., number of nodes, connectance, nestedness, modularity), illustrated by the principal component analysis (along a PC1 axis explaining 70% of the variance, Fig. 3a). Across all 927 food webs, the blue food webs were in general smaller (median number of nodes in blue: 35; in green: 437), more connected (median connectance in blue: 0.25; in green: 0.06), and less modular (median modularity in blue: 0.03; in green: 0.20) than the green ones (Fig. 3a).

**Associations between food webs and environmental drivers**
We then analysed the associations between properties of local food webs and the focal environmental drivers, namely elevation and land use. We focused on elevation over temperature since mechanistically the two are highly co-linear, and the effects of temperature were encapsulated within those of elevation (Supplementary Table 4). By considering the mutual dependence among food-web metrics using piecewise structural equation modelling (SEM) analyses, we found elevation to be significantly associated with most of the structural and ecological (i.e., diet niche overlap) properties in both blue and green food webs, but with contrasting relationships (Fig. 3b, c). Regarding

direct effects, elevation positively influenced modularity (standardised coefficient: 0.49) and negatively influenced niche overlap (−0.53) in green food webs, while the opposite (−0.38 and marginally significant 0.10, respectively) was observed in blue ones (Fig. 3c; Supplementary Table 7). Further SEM analyses separated by land-use types with subsetted food webs detected the same set of contrasting relationships between the two systems, particularly in farmlands (Supplementary Fig. 1; Supplementary Tables 8–12). General linear model analyses supported that elevation and local dominant land-use type are both significant drivers of food-web properties (Supplementary Table 3). Taken together, the results suggest that blue and green food webs change differently in their structural and ecological properties along elevation, resulting from each of their elevational community composition turnover, and such blue-green differences were most obvious in farmlands.

Next, to investigate potential nonlinear elevational patterns, we analysed food-web properties along elevation with generalised additive models considering all 972 food webs. In addition, to identify potential different responses among land-use types, we also conducted similar analyses with linear models comparing regression

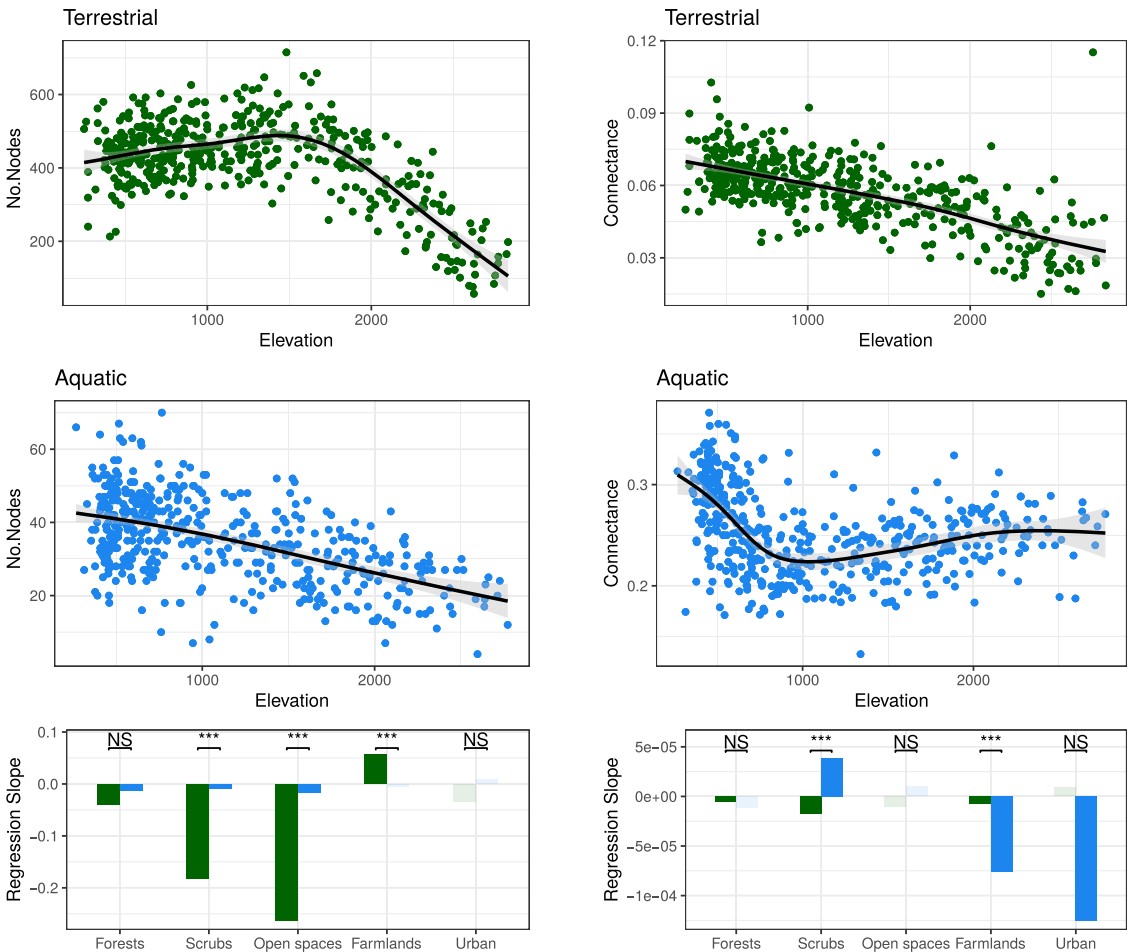

**Fig. 4 | Number of nodes (left panel) and connectance (right panel) of terrestrial (green) and aquatic (blue) food webs along elevation.** In the scatter plots, black lines and corresponding shades are the fitted regression and 95% CI of generalised additive models. The bottom barplots present the slope estimates from linear models testing the effects of elevation on the focal food-web property with subsetted food webs of each dominant land-use type. Solid green/blue bars indicate significant slopes, likewise faded colours non-significant ones. The significance of terrestrial versus aquatic least-squares slope comparison (two-tailed) is indicated above the bars (***P < 0.001, NS non-significant; details in Supplementary Fig. 3).

slopes of food webs subsetted by land-use types. Regarding the number of nodes (i.e., species richness), the green food webs showed an abrupt changing pattern with elevation, as the number of nodes increased with elevation until about 1500–2000 m a.s.l. but decreased thereafter (Fig. 4). This abrupt change coincides with the tree-line effect on community composition[33,34]. The blue food webs, conversely, showed a consistent linear decrease with elevation. Further blue-green slope comparisons by land-use types revealed that the main qualitative difference between the patterns in the two systems (which occurred below 1500 m a.s.l.) was largely due to their different responses to elevation in farmlands (Fig. 4; Supplementary Fig. 3): green food webs became larger because more plants and butterflies were added, while blue ones became smaller because more fishes were lost than the added invertebrates (Supplementary Figs. 6–11). The connectance decreased near linearly with increasing elevation in green food webs, whereas in blue ones connectance decreased relatively quickly with increasing elevation until 1000 m a.s.l. but mildly increased above 1000 m a.s.l. Blue food webs became less connected with increasing elevation due to the gradual loss of fishes, and thus the many trophic links toward the overall more-generalist fishes' diets, up to roughly 1000 m a.s.l. (Supplementary Fig. 11). At above 1000 m a.s.l., fish became very rare and invertebrate the dominant group (Supplementary Figs. 10 and 11). The increasing food-web connectance with increasing elevation thus reflected the combined effect of the

shrinking food-web size (Fig. 4) and the replacement of specialist by generalist invertebrates (Supplementary Fig. 10; *sensu*[35]).

For nestedness, modularity, and diet niche overlap, we further analysed the inferred local food webs against two types of randomisation (i.e., keep-group and fully randomised webs; see *Methods*), to understand whether the observed patterns were driven by the change in food-web size and connectance, or by the change in the composition of focal group or species (and thus the corresponding diet composition) along elevation. In general, green food webs were more nested and more modular (when below 2500 m a.s.l.) than their randomised counterparts (Fig. 5). Conversely, blue food webs were more nested but less modular than their randomised counterparts (Fig. 5). Both blue and green food webs showed a trend of decreasing nestedness with increasing elevation (Fig. 5). In the blue food webs, specifically, the decrease in nestedness occurred exclusively until an elevation of 1000 m a.s.l. was reached, after which nestedness plateaued (Fig. 5; Supplementary Fig. 4). This rapid drop of nestedness in blue food webs again (as in connectance) matched the "fish-line" effect, that is, fish species gradually dropped out with increasing elevation, and only very few species remained above 1000 m a.s.l. (Supplementary Fig. 11; see *Discussion*). Loss of fish species richness strongly influenced food-web nestedness not only because it reduced connectance (captured by fully randomised webs, Fig. 5), but also because most fish species were generalist consumers (with much broader diets than even generalist

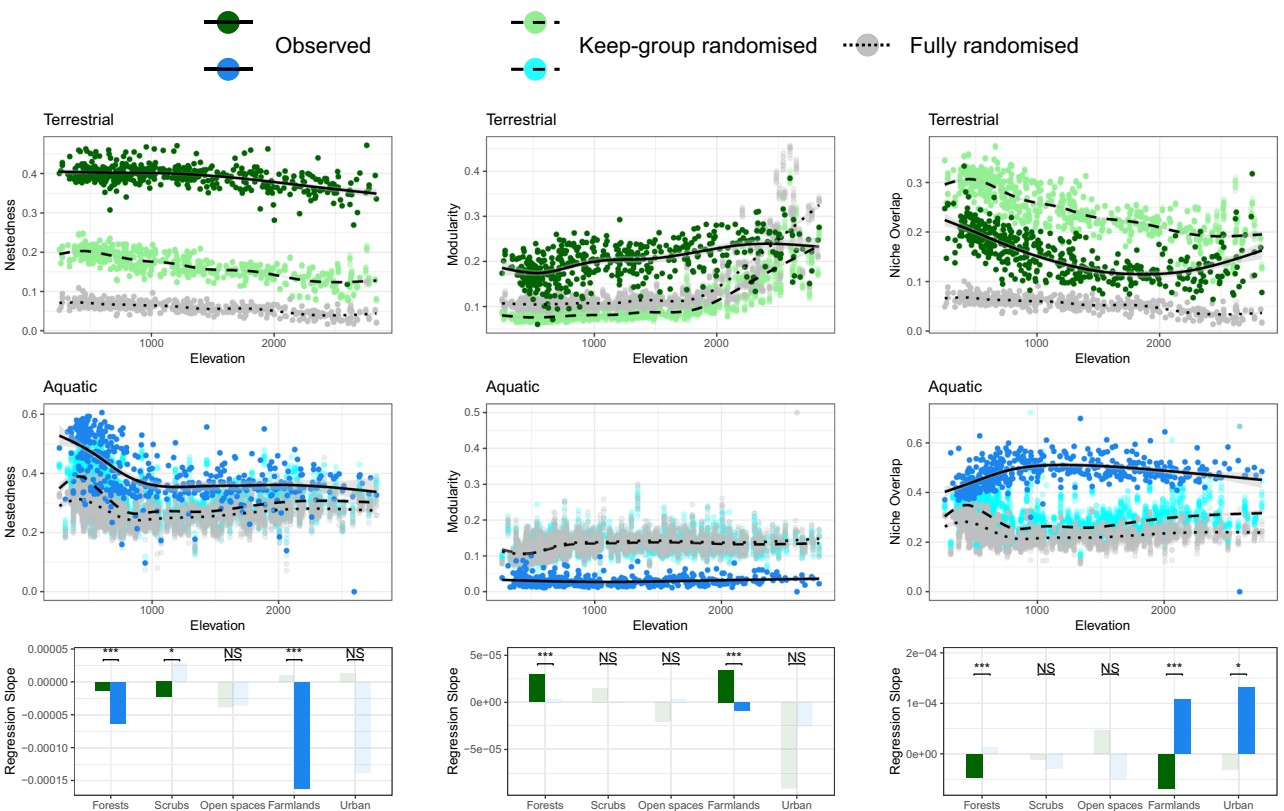

**Fig. 5 | Nestedness (left panel), modularity (middle panel), and consumers' niche overlap (right panel) of terrestrial (green) and aquatic (blue) food webs along elevation.** In the scatter plots, besides the observed values from the inferred food webs (green/blue), the values from the keep-group (light green/blue) and fully (grey) randomised counterparts are presented alongside for comparison. The black lines (solid, dashed, and dotted) are the fitted regression of generalised additive models with corresponding shades the 95% CI. The bottom barplots present the slope estimates from linear models testing the effects of elevation on the focal food-web property with subsetted food webs of each dominant land-use type. Solid green/blue bars indicate significant slopes, likewise faded colours non-significant ones. The significance of terrestrial versus aquatic least-squares slope comparison (two-tailed) is indicated above the bars (*$P < 0.05$, ***$P < 0.001$, NS non-significant; details in Supplementary Fig. 4).

invertebrates, see Supplementary Figs. 10 and 11) that tended to shape a nested structure in food webs (captured by keep-group randomised and inferred webs, Fig. 5). The green food webs, but not the blue ones, saw a mild increase of modularity with elevation (Fig. 5). At lower elevation, such an increase was due to a boost of richness with elevation in butterflies, but not grasshoppers and birds (Supplementary Figs. 6–9), leading to a higher proportion of specialist consumers within communities. However, this association between butterfly richness and elevation was inversed beyond a turning point around the tree line (Supplementary Fig. 7). Thereafter, as the influences of decreasing specialist proportion (disfavouring modularity) and decreasing food-web size and connectance (favouring modularity, as shown in randomised webs in Fig. 5) could have balanced out each other, the observed modularity of green food webs became more or less unchanged (Fig. 5; Supplementary Fig. 4).

The consumers' diet niche overlapped more in inferred food webs than in fully randomised webs in both blue and green systems (Fig. 5), but the green food webs overlapped less while the blue ones overlapped more than their respective keep-group randomised counterparts (Fig. 5). These patterns showed that, within each of the focal consumer groups (i.e., excluding plants and aquatic basal resources), species-specific diets are more differentiated among terrestrial consumers, while more overlapped among aquatic ones, compared to expectation by chance. Both types of food webs had nonlinear patterns of niche overlap along elevation, but in different ways: the green food webs saw a decrease until about 1500–2000 m a.s.l., and switched to an increase thereafter (Fig. 5), again reflecting the effects of the tree

line and the change in the richness of specialist butterflies (Supplementary Fig. 7). Contrastingly, the blue food webs saw an increase in niche overlap until about 1000 m a.s.l. then remained roughly constant thereafter (Fig. 5). This increasing niche overlap together with the decreasing nestedness along elevation up to 1000 m a.s.l. reflected that, as fish species richness decreased along elevation, it was those relatively-generalist invertebrates sharing alike diets that became the majority in stream communities (Supplementary Fig. 10) and determined food-web structure.

For all five food-web metrics analysed against elevation, we found significant differences in blue versus green regression slopes in farmlands (Figs. 4 and 5; Supplementary Figs. 3 and 4). There specifically, the blue and green slopes were both statistically significant and having opposite signs in two metrics, such that modularity was positively and diet niche overlap negatively associated with elevation in green food webs, and the inverse in blue ones (Figs. 4 and 5; Supplementary Figs. 3 and 4). These were the most pronounced blue-green differences we detected among all land-use types. In accordance with the findings from our SEM analyses by land-use type, in general, it was especially in farmlands where the blue and green food webs exhibited qualitatively different responses to elevation.

## Discussion
Examining ecological communities from the perspective of not only the richness of species, but also of their interaction networks can inform the management of biodiversity and ecosystem functioning[1,7]. We showed that local food webs exhibit variation in their structural

and ecological properties corresponding to a change in elevation and variation in surrounding land-use types. Specifically, terrestrial (green) and aquatic freshwater (blue) food webs responded differently to these environmental factors. This suggests that the same environmental change can influence terrestrial and aquatic taxa differently, thereby imposing complicated overall impacts on the biodiversity and functions of local ecosystems (*sensu*[36]). As environmental gradients are reshaped by global changes, the complex responses of food webs challenge the development of effective management and conservation strategies.

Using nearly a thousand inferred local food webs representatively covering Switzerland, we saw clear and gradual changes in almost all key food-web metrics with elevation in both blue and green systems. In many cases, the overall pattern was nonlinear. Some of such nonlinear food-web responses along elevation were driven by bio-geographical boundaries, e.g., the tree line (around 1500–2000 m a.s.l. across our local sites, see Fig. 2) in green food webs, while some were associated with food-web responses specific to land-use types in combination with a land cover turnover by elevation (Fig. 2; Supplementary Figs. 3–5). The elevation gradient is often taken as a proxy of climatic change in ecological research (e.g., ref. 37), and important climatic variables, such as temperature and precipitation, change consistently along elevation. Indeed, the herein-considered elevation gradient encapsulated, among others, effects of temperature on food webs (revealed by temperature-based analyses, see *Methods* and Supplementary Table 4). Thus, by looking at elevation, we captured most of the temperature influences meanwhile still included those of other non-temperature elevational drivers on food webs. In terrestrial systems, our focal consumers, i.e., butterflies, grasshoppers, and birds, are mostly highly mobile. Their elevational distributions are thus expected to be influenced not so much by topographical constraints as by their physiological constraints in relation to temperature and bottom-up reliance on temperature-shaped vegetation[38–40]. Therefore, the green food-web patterns we observed along elevation likely reflected the effects of elevational temperature variation on species distribution and thus the composition of communities. In contrast, in stream systems, the existence of many taxa is also constrained by the steepness (thus flow speed) or width of the stream, which both covariate with elevation[41,42]. For instance, only very few fish species, mostly brown trout, are present above 1000 m a.s.l. (i.e., the "fish line", Supplementary Fig. 11) because the structure and hydrology of streams does not favour or allow many fishes to occur at these elevations. Indeed, we saw a fish-line effect in several detected elevational patterns of our blue food webs (Figs. 4 and 5). The temperature effects are nonetheless still identifiable, as the observed patterns below the fish line were broadly in line with temperature-driven ones reported in the literature, such that higher temperature favours the presence of a larger number of fish species in streams[43] (Supplementary Fig. 11; especially of the predominant members of Cyprinidae) and higher food-web connectance[44] (Fig. 4). Overall, the blue food-web patterns along elevation are likely shaped simultaneously by both temperature and topographical effects, as elevation and temperature are less strongly correlated in aquatic versus terrestrial systems, and topography and hydrology are per se strong drivers in the former.

The nested and non-modular structure detected in the blue food webs (Fig. 5) well-echoes the reports in the literature[45,46], reflecting that aquatic consumers are usually broad feeders with diets mostly constrained by body size (or say gape size) instead of other feeding traits[47,48]. In contrast, the relatively high modularity observed in the green food webs (Fig. 5) suggests a relatively high proportion of specialised consumer-resource pairs within communities[49] (Supplementary Fig. 7). This is further supported by the fact that green food webs had lower while the blue ones had higher diet niche overlap than their respective keep-group randomised counterparts (Fig. 5). It is an inherent property of aquatic and terrestrial food webs, respectively, that the latter contains a very large part of highly specialised co-evolved interaction partners (i.e., plant-insect co-evolution), while the former are more constrained by resource availability[48]. Therefore, diet niche differentiation versus overlap may not be as important a constraining factor for species coexistence in stream communities as in terrestrial ones. Interestingly, in terms of diet specialisation and differentiation, elevation had significant yet opposite influences on blue and green food webs, especially in farmlands (Figs. 3c, 5; Supplementary Figs. 1, 4). With increasing elevation and thus decreasing temperature, green food webs were increasingly composed by more-specialist butterflies and birds whose diets were more differentiated, whereas blue food webs by more-generalist fishes and invertebrates whose diets were more overlapped (Supplementary Figs. 6–11). These findings thus suggest different elevation-dependent strategies for managing local biodiversity between the two systems, yet also indicate that management strategies applied to terrestrial systems may have opposing cascading effects on aquatic systems[50]. Toward higher elevation, we should emphasise more on keeping diverse resources in green communities to support the living of specialists, meanwhile stabilising resource quantities in blue communities, given the high diet niche overlap can impose excessive competition among consumers when the shared resources become rare[51,52]. In addition, with a global-change prospect, climatic warming can threaten biodiversity via imposing physiochemical conditions that no longer suits local populations[53], or triggering biological range shift where new biological interactions emerge[54]. According to our findings, terrestrial communities may be especially vulnerable to the latter mechanism, such that warming can spark an elevational upshift of generalist species into regions originally dominated by specialists, provoking predation or resource competition pressures and thus increasing threat to the latter. Importantly, however, elevational shifts of species may be limited by other factors (such as hydrology, topography) than local temperature only, and may affect different organismal groups differently.

Anthropogenic land use has been shown to be a major driver of the richness and community composition of different taxonomic and trophic groups (e.g., refs. 36, 55), but rarely associated with food-web properties (e.g., refs. 13, 56), and here compared between blue and green systems. We spotted notable land-use effects as (i) the different elevational patterns between blue and green food webs are mostly contributed by webs in farmlands, and (ii) within the green system, food webs in farmlands and forests adopt different elevational patterns. With regard to the blue-green difference, our focal land-use types are defined based on the vegetation or human modifications on land, which often also cascade to aquatic systems[50], but are not describing the streams themselves that are embedded into the terrestrial matrix. Nonetheless, we detect significant land-type effects not only in green but also in blue food webs (Figs. 4 and 5; Supplementary Figs. 3 and 4). There are various interchanges between terrestrial and aquatic systems that may provide channels for potential cross-ecosystem spill-over effects on community composition[57]. For instance, leaves and terrestrial organisms become the inputs of organic matter to water bodies, and many aquatic insects at their late life stages emerge into terrestrial communities[58,59]. However, in our case, the comparison among blue food webs with controlled elevation revealed no obvious difference between forests and farmlands (Supplementary Fig. 5), indicating that the detected land-use effects in blue food webs reflect mostly their elevational responses in combination with changes in land-use type along elevation, rather than spill-over effects from land. With regard to the green food webs among land-use types, webs in farmlands tended to be smaller and have higher overlap in diet niche than those in forests at low elevation, but not at higher elevation (Supplementary Fig. 5). Compared to natural conditions, agriculture generally simplifies local vegetation into a few selected crops and exerts frequent perturbations (e.g., harvesting) on the habitat, leading to conditions that favour generalists over specialists[60].

Indeed, at low-elevation farmlands where agricultural intensity is relatively high, the terrestrial communities fostered fewer plants, and fewer but more-generalist butterflies, than at either high-elevation farmlands or low-elevation forests (Supplementary Figs. 6 and 7). The different elevational food-web patterns we observed in forest versus farmland green food webs suggest that anthropogenic land use and climate change can have interactive ecological impacts. The warming effects on food webs can depend on human-caused habitat modifications and its intensity, such as those caused by agricultural activities, given farmlands consistently being a pronounced and outstanding type in our land-use-relevant discoveries. Such effects, as detected by contemporary species occurrences and inferred food-web structures, could potentially be a result of both contemporary environmental and biotic drivers, as well as of past drivers leaving a legacy.

The empirical observation-based taxa occurrence and trophic interaction information allowed us to examine comparatively realistic food webs across a large spatial scale, without the necessity to base the local food-web construction on probabilistic calculations [e.g., ref. 32]. While there are further species (and more taxonomic groups) occurring in these local communities, our current taxa coverage is extensive, covering a broad range of taxonomic and functional groups that are of high importance in these ecosystems as herbivores, predators or primary producers. We could not include less-well studied taxonomic groups due to the lack of detailed information on their occurrence or/ and trophic interactions. For the same reason, we simplified the basal resources in the aquatic system as three mega nodes without detailed identities (see *Methods*). These were compromises made between the resolution (and reliance) of compiled empirical knowledge and its coverage that we needed to embrace. Future work may broaden the taxa coverage once detailed data become available. Importantly, because these simplifications were consistent across all the local sites we studied, the detected intra-biome spatial patterns were comparable and relied on the same subset of the true complete food web. This also enabled our blue-green comparisons, with which our biological interpretations emphasised the qualitative pattern-wise differences instead of the quantitative metric-wise ones between the two biomes. Finally, freshwaters are embedded in the terrestrial matrix after all. While we here reveal the spatial structural patterns of blue and green food webs separately, a blue-green synchronous sampling in an interconnected framework could potentially allow further investigations of cross-system population dynamics at a relatively finer temporal resolution (see our structural exploration and relevant discussion in Supplementary Discussion).

In conclusion, our large-scale analysis of food-web structure and change in blue and green systems provides evidence that blue and green food webs within the same landscape respond differently (in terms of major network metrics) to elevation and land use. The food-web patterns featured in this study emerge spatially from community compositional differences, which supposedly reflect the outcome of both evolutionary and ecological processes. By making analogies between elevation and climate change, our findings provide not only a broad depiction of both blue and green food webs with their current status across the landscape, but also visionary implications with their potential future change. Such understandings could become fundamental knowledge when managing local biodiversity and ecosystem functioning, especially in places where blue and green communities coexist and are vulnerable to anthropogenic modifications.

## Methods

### Overview
We compiled systematically sampled empirical taxa occurrence across the landscape, and inferentially assembled respective blue and green local food webs by combining these data with a metaweb approach. We quantified key properties of the inferred food webs, then analysed with GIS-derived environmental information how focal food-web

metrics change along elevation and among different land-use types in blue versus green systems. Details are given below.

### Assemble food webs using a metaweb approach
We applied a metaweb method to obtain the composition and structure of multiple local food webs across a landscape spatial scale[10]. A metaweb is an accumulation of all interactions (here, trophic relationships) among the focal taxa. In this study, we built our metaweb based on known trophic interactions derived from literature and published datasets, which themselves were all based on primary empirical natural history observations. We further complemented or refined the trophic interactions in the metaweb based on expert knowledge of primary observations that are not yet published or only accessible in grey literature. The expert knowledge covers authors and collaborators who have specific natural history knowledge on Central European plants, herbivorous insects, birds, fish, and aquatic invertebrates. Importantly, these observations were all based on empirical observations and/or unpublished data accumulated over considerable field research experience. The respective literature we referred, as well as the metaweb itself with information source of each trophic link (online repository), are provided in Supplementary Methods. By assuming that any interaction in the metaweb will realise if the interacting taxa co-occur, the metaweb approach allows an inference of local food webs if taxa occurrence is known. Such an assumption of fixed diets may lead to an over-estimation of the locally realised trophic links[32], as it essentially ignores the possible intraspecific diet variation caused by resource availability[61,62], predation risk[63], temperature[64], ontogenetic shift[65], or other genetic and environmental sources[66]. Therefore, the food webs we inferred systematically using this method capture trophic relationships driven by community composition (species presence versus absence) but not the above-mentioned processes. Nonetheless, since the trophic interactions were based on empirical observations, the fixed diets can be seen as collapsing all intraspecific variations of diet-determining traits (or trait-matching) at species level, within which we know realisable interactions surely exist. This, together with co-occurrence as a pre-requisite, gives realistic boundaries for the potential interaction realisation, which is plausible and non-biased when applying to localised sites. With this approach, we were addressing a systematic comparison among potential local food webs between the blue and green systems and across the selected gradients. For sensitivity analyses considering the potential inaccuracy of the metaweb approach mentioned here, please see further below *Food-web metrics and analyses* and Supplementary Discussion.

We compiled taxa occurrence of four terrestrial and two aquatic broad taxonomic groups ("focal groups") to assemble local green and blue communities, respectively and independently, based on the well-resolved data available. Each focal group referred to a distinct taxonomic group, and the within- and among-group trophic relationships captured most of the realised interactions. These focal groups were vascular plants, butterflies, grasshoppers, and birds in the green biome, and stream invertebrates and fishes in the blue biome. Notably, with "butterflies" we refer to their larval stage and accordingly their mostly-herbivorous trophic interactions throughout this study. Larval interactions were also the predominant interaction assessed for stream invertebrates (i.e., all interactions of stream invertebrates focussed on their aquatic stage, which is predominant larval). The occurrence data of these focal groups were compiled from highly standardised multiple-year empirical surveys of various authorities, all conducted by trained biologists with fixed protocols (Supplementary Methods). The information across sites should thus be representative and can be up-scaled to the landscape. The occurrences of plants, butterflies, birds, and stream invertebrates were from the *Biodiversity Monitoring Switzerland* programme (BDM Coordination Office[67]) managed by the Swiss Federal Office for the Environment (BAFU/

FOEN). The occurrences of grasshoppers and fishes were from the Swiss database on faunistic records, *info fauna* (CSCF), where we further complemented fish occurrence from the data of *Progetto Fiumi Project* (Eawag). In terms of biological resolution, taxa were resolved to species level in most cases, while the plant and butterfly groups included some multi-species complexes. Insects of the order Ephemeroptera, Plecoptera, and Trichoptera were resolved to species, while all other stream invertebrates were resolved to family level. These were each treated as a node later in our food-web assembly, and referred to as "species", as the species within such complexes and families mostly share the same trophic role. Spatially, the occurrence datasets adopted coordinates resolved to $1 \times 1\,km^2$. The species that were recorded in the same $1 \times 1\,km^2$ grid were considered to co-occurred. We took the co-occurring four/two focal groups to form local green/blue local communities, respectively. To obtain better co-occurrence across group-specific data from different sources (e.g., *BDM* and *info fauna*), we intentionally coarsened the grasshopper and fish occurrence to $5 \times 5\,km^2$ coordinates. This is arguably a biologically acceptable approximation considering the high mobility of these two groups. Also, we only included known stream-borne fishes and dropped pure lake-borne ones to match our stream-only invertebrate occurrence data. Across all 462 green and 465 blue communities we assembled, we covered 2016 plant, 191 butterfly, 109 grasshopper, 155 bird, 248 stream invertebrate, and 78 stream fish species. Unlike the knowledge of plant occurrence in green communities, we did not have detailed occurrence information of the basal components (e.g., primary producers) in blue ones. Therefore, we assumed three mega nodes—namely plant (including all alive or dead plant materials), plankton (including zooplankton, phytoplankton, and other algae), and detritus—as the basal nodes occurring in all blue communities, without further discrimination of identities or biology within. These adding to our focal groups thus cover major taxonomic groups as well as trophic roles from producers to top consumers in both blue and green systems.

Taking the above-assembled local communities then drawing trophic links among species (nodes) according to the metaweb yielded the local food webs (illustrated in Fig. 1), representatively covering the whole Swiss area. Notably, although our understanding of trophic interactions indeed encompassed some links across the blue and green taxa (e.g., between piscivorous birds and fishes), our occurrence datasets did not present sufficient spatial grids where these taxa co-occur. We, therefore, did not include such links, nor assembled blue-green interconnected food webs, but the blue and green food webs separately instead (but see Supplementary Discussion). Also, we dropped isolated nodes, i.e., basal nodes without any co-occurring consumer and consumer nodes without any co-occurring resource, from the inferred food webs. These could possibly be passing-by species that were recorded but had no trophic interaction locally, or those that interact with non-focal taxa whose occurrence information was unknown to us. We thus had to exclude them to focus on evidence-supported occurrences and trophic interactions. Nonetheless, across all cases, isolated nodes were rather rare (averaged less than 3% of species occurred in either blue or green communities).

### Environmental data
We acquired environmental data across all of Switzerland (42,000 $km^2$) on a $1 \times 1\,km^2$ grid basis (i.e., values are averaged over the grid) from GIS databases, with which we mapped environmental conditions to the grids where we assembled food webs. These included: topographical information from *DHM25* (Swisstopo, FOT), land-cover information from *CLC* (EEA), and climate information (averaged over the decade of 2005–2015) from *CHELSA*. Among environmental variables, elevation and temperature are essentially highly correlated. In this study, we took elevation as the focal environmental gradient throughout, as after accounting for the main effects of

elevation on temperature, the residual temperature was not a good predictor of the food-web metrics we looked at (see next section, and Supplementary Table 4). In other words, by analysing along the elevation gradient, we already captured most of the temperature influences on food webs. Based on the labels provided by the GIS databases, we categorised the originally detailed land cover into the five major land-use types that we used in this study, namely forest, scrubland, open space, farmland, and urban area. Forest includes broad-leaved, coniferous, and mixed forests. Scrub includes bushy and herbaceous vegetation, heathlands, and natural grasslands. Open space encompasses sparsely vegetated areas, such as dunes, bare rocks, glaciers and perpetual snow. Farmland include any form of arable, pastures, and agro-forestry areas. Finally, urban area is where artificial constructions and infrastructure prevail. As each grid could contain multiple land-use types, we then defined the dominant land-use type of the grid as any of the five above that occupied more than 50% of the grid's area. Analyses separated by land-use types with subsetted food webs (land-use-specific analyses) were based on the grids' dominant land-use type. There were a few grids where the dominant land-use type did not belong to the focal major five, e.g., wetlands or water bodies, and a few where no single land-use type covered more than 50% of the area. Food webs of these grids were still included in the overall analyses but excluded from any land-use-specific analyses (as revealed in the difference in sample sizes between all versus land-use type subsetted food webs in Fig. 2; analyses details below).

### Food-web metrics and analyses
We quantified five metrics as the measures of the food webs' structural and ecological properties. For the fundamental structure of the food webs, the number of nodes ("No. Nodes") reflects the size of the web, meanwhile represents local species richness (though the few isolated nodes were excluded as above-mentioned). Connectance is the proportion of realised links among all potential ones (thus bounded 0–1), reflecting how connected the web is. We also derived holistic topological measures, namely nestedness and modularity. Nestedness of a food web, on the one hand, describes the tendency that some nodes' narrower diets being subsets of other's broader diets. We adopted a recently developed UNODF index[68] (bounded 0–1) that is especially suitable for quantifying such a feature in our unipartite food webs. On the other hand, modularity (bounded 0–1 with our index) reflects the tendency of a food web to form modules, where nodes are highly connected within but only loosely connected between. Nestedness and modularity are two commonly investigated structures in ecological networks and have been considered relevant to species feeding ecology[24] and the stability of the system[69]. Finally, we measured the level of consumers' diet niche overlap of the food webs (Horn's index[70], bounded 0–1), which essentially depends on the arrangement of trophic relationships (thus the structure of the webs), and could have strong ecological implications as niche partitioning has been recognised to be a key mechanism that drives species coexistence[71,72]. We selected these fundamental and holistic properties as they are potentially more relevant to the processes that may have shaped food webs across a landscape scale (e.g., community assembly), in comparison to some node- or link-centric properties. Also, addressing similar metrics as in the literature[13,69] would facilitate potential cross-study comparison or validation.

To first gain a glimpse of the structure of the blue and green food webs, we performed a principal component analysis (PCA; Fig. 3a) on the inferred food webs ($n = 462$ and 465 in green and blue, respectively) taking the four structural metrics (number of nodes, connectance, nestedness, and modularity) as the explaining variables of blue versus green system types. We then confirmed that system type, elevation, and land-use type were all important predictors of food-web metrics (whereas the residual temperature after accounting elevation

effects was not) by conducting general linear model analyses, taking the former as interactive predictors while the latter response variables (Supplementary Tables 3, 4). To check how elevation influences food-web properties in blue and green systems separately, and how food-web properties depend on each other, we ran a series of piecewise structural equation modelling (SEM)[73] analyses on inferred food webs (Fig. 3b, c) whose dominant land use can be defined ($n = 421$ and 430 in green and blue, respectively). This was also conducted on subsetted webs of each of the five major land-use types (Supplementary Figs. 1 and 2). The SEM relationships were derived from linear mixed model analyses with dominant land-use type as a random effect (assumption tests see Supplementary Figs. 12–17). The SEM structure of direct effects was set according to the literature[13,69] and is illustrated in Fig. 3b. In short, this structure tests the dependencies from elevation (an environmental predictor) to food-web metrics (ecological responses). The further dependencies among food-web metrics themselves were assigned with the principle of pointing from relative lower-level properties to higher-level ones. That is, from number of nodes (purely determined by nodes) to connectance (determined by numbers of nodes and links), further to nestedness and modularity (holistic topologies, determined further by the arrangement of links), then to diet niche overlap (ecological functional outcome).

Finally, to check and visualise the exact changing patterns of food webs, we applied generalised additive models (GAMs) to reveal the relationships between food-web metrics and the whole-ranged elevation (Figs. 4 and 5), as well as a particular comparison between food webs in forests and farmlands below 1500 m a.s.l. (Supplementary Fig. 5), as this elevation segment covered most of the sites belonged to these two land-use types. We also performed a series of linear models (LMs) and least-squared slope comparisons based on land-use-specific subsets of food webs (Figs. 4 and 5; Supplementary Figs. 3 and 4), to investigate whether food-web elevational patterns are different among land-use types (assumption tests see Supplementary Tables 5 and 6). In the GAMs analyses, specifically, we simulated two sets of randomised webs, i.e., "keep-group" and "fully", as the null models to compare with the inferred ones[74]. Both randomisations generated ten independently simulated webs from each input inferred local food web, keeping the same number of nodes and connectance as of the latter. On the one hand, the keep-group randomisation shuffled trophic links from an input local web but only allowed them to realised fulfilling some pre-set within- and among-group relationships. That is, in green communities, birds can feed on all groups, grasshoppers on any groups but birds, while butterflies only on plants; in blue communities, fishes can feed on all groups, while invertebrates on themselves and the basal resources. These pre-set group-wide relationships captured the majority of realistic trophic interactions compiled in our metaweb. On the other hand, the fully randomised webs shuffled trophic links disregarding the biological identity of nodes. The GAMs of nestedness, modularity, and niche overlap illustrated the patterns of these randomised webs (Fig. 5). Comparing among the three types of webs, the patterns exhibited already by fully randomised webs should be those contributed by variations in web size and connectance, while the difference between keep-group and fully randomised webs by the focal-group composition of local communities, and the difference between inferred and keep-group randomised webs further by the realistic species-specific diets. In addition, we also applied the same GAMs and LMs approach to analyse node richness, as well as both realised and potential diet generality (vulnerability for plants) of each focal group (Supplementary Figs. 6–11). These analyses provided hints about the changes in community composition and species diet breadths along elevation and among land-use types, which helped explain the detected food-web responses in mechanistic ways.

In addition, to check if our findings were shaped or strongly influenced by the potential inaccuracy of using the metaweb, we repeated the above PCA, SEM, and GAM analyses as a series of sensitivity analyses. We generated food webs based on our locally inferred ones (i.e., the observations) but with random 10% link removal. This procedure mimics the effect of potential intraspecific diet variation (mentioned earlier) so that some trophic interactions in the metaweb do not realise locally. Overall, these analyses with link removal showed that our conclusions are qualitatively and quantitatively highly robust, and only very minorly affected by the such potential inaccuracy of metawebs, which is also in accordance to other food-web studies (see e.g., Pearse & Altermatt 2015[75]). All details and outcomes of these additional analyses are given in Supplementary discussion.

All metric quantification and analyses were performed under R version 4.0.3 (R Core Team[76]). All applied packages and functions were described in Supplementary Methods, while the R scripts performing these tasks can be accessed at the online repository provided.

### Reporting summary
Further information on research design is available in the Nature Research Reporting Summary linked to this article.

## Data availability
Taxa occurrence (from databases/datasets of BDM: https://www.biodiversitymonitoring.ch/, *info fauna*: http://www.cscf.ch/cscf/, and *Progetto Fiumi* Project of Eawag) and GIS environmental information (from databases including *DHM25*: https://www.swisstopo.admin.ch/en/geodata/height/dhm25.html, CLC: https://land.copernicus.eu/pan-european/corine-land-cover, and *CHELSA*: http://chelsa-climate.org) are data that we obtained from respective authorities in charge (as listed in the *Methods* section), which can be accessed by contacting these authorities. The metaweb trophic-interaction data and the processed local food-web data underlying the figures and tables in this study are available at a public repository here: https://doi.org/10.6084/m9.figshare.17817152.v1.

## Code availability
The R codes that reproduce all analyses and figures in this study are provided at a public repository: https://doi.org/10.6084/m9.figshare.17817152.v1.

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

## Acknowledgements

We thank the Swiss Federal Office of the Environment (FOEN) and the Coordination Office of the Biodiversity Monitoring Switzerland (*BDM*, especially Tobias Roth) for providing plant, butterfly, and bird occurrence data from the *Schweizerische Vogelwarte* and the *BDM*, as well as the Centre Suisse de Cartographie de la Faune (CSCF) and Yves Gonseth for providing grasshopper and fish occurrence data from the *info fauna*. We thank Rosi Siber for extracting the needed GIS data. We thank Raffael Ayé, Thomas Sattler, Felix Neff, Luiz Jardim De Queiroz, and Carmela Dönz for their respective assistance and input in compiling trophic relationships among the focus groups in this study. Funding is through the ETH Board Blue-Green Biodiversity (BGB) Initiative (BGB2020), the Swiss National Science Foundation Grant # 310030_197410 and # 31003A_173074, and the University of Zurich Research Priority Programme in Global Change and Biodiversity (URPP GCB) to Florian Altermatt.

## Author contributions

F.A. and L.P., together with J.B., M.G., C.H., O.S. and N.Z., developed the idea and secured the funding. H.H., together with F.A., S.K. and M.R.C., compiled the data. H.H. conducted the analyses and drafted the manuscript with substantial inputs by L.P. and F.A. All authors contributed to subsequent revisions of the manuscript.

## Competing interests

The authors declare no competing interests.
