## [Peer Review File · Nature Communications]

Blue and green food webs respond differently to elevation and land useReviewer #1 (Remarks to the Author):

Summary

Biodiversity is under threat from global changes and there is a pressing need to predict how community will respond to these changes. A key part of biological communities is the interactions that bind species yet we have only recently begun to study how these interactions are changing. The few studies that investigate this problem focus on bipartite webs in one ecosystem. In this contribution, the authors assemble an outstanding data set on food webs in terrestrial (green) and aquatic (blue) ecosystems to investigate how elevation, climate and land-use impact food web properties. The authors offer a tantalizing set of results that demonstrate key drivers of food web variation both within ecosystem (i.e., across blue webs or across green webs) and across ecosystems (i.e., blue vs green webs).

General comments:

1. This is an impressive study! The major strengths are the strong data sets, robust analysis of both within and across ecosystem drivers of food web change, and insightful interpretations that will make this study a foundation for future work. The authors have identified a clear gap and developed a strong study to fill this gap.
2. At first, I thought the authors were going to present full connected green and blue webs – so I was a bit disappointed that this was not done. That said, I fully value the approach taken and see it as a logical first step. But I want to know more about the potential connections across green-blue webs. Are there really few sampling sites that accommodated these potential connections. In this system, is it possible that the species that connect green and blue may be critical species to consider for landscape management?
3. The data are super impressive – wow. But clearly these data are only a subset of the webs – particularly in terrestrial systems. Do we know anything about how higher level species (ie predators) are distributed across the terrestrial gradients? I was surprised not to see any discussion of this.
 - a) The environment and species are changing. I would like to know how the timing of sampling of the different biotic and abiotic data compare here. There is some mention of the timing of some components of the abiotic side but the reader needs to dig into the references to get more information on most other pieces of data. I recommend the authors consider adding a table with sampling period in appendix. If sampling of biotic occurs in a very different time frame than climate data, then the authors might consider constraining their analysis to data that are sampled in the same period. Otherwise, I wonder if variation in sampling period might be adding some noise to these relationships.
 - b) Do the authors have replicates in the sampling of blue webs – ie two sampling sites on one stream? If so, did the authors treat this nested structure in their analyses?
 - c) lines 348-350. How does making this assumption influence your results – this sets a strict number of nodes as basal resources so will constrain your metrics. It is still a small number relative to the total number of nodes but might be worth a discussion (in appendix?). I would expect higher connectance and lower modularity in blue vs green if basal resources are unchanging in blue. And I believe this is what you found. This would not affect intraspecific (ie blue vs blue OR green vs green) web comparisons but maybe interspecific (ie blue vs green).
 - d) lines 394-408, Figure 3B. I recommend the authors walk the reader through Fig 3B in appendix. I think this will help make the work more accessible to a broad audience. There is a lot under the hood here! The authors are use common food web metrics. Did the authors have any a priori predictions for relationships or was this purely exploratory?
 - d) lines 434-444 This is a nice explanation of the value of the two randomized webs here.
4. The discussion is very well done – it blends specific results with broad interpretations (ie beyond their system). The land-use type analysis is quite fascinating and rare at this scale. But I am wondering about history or legacy effects here. To what extent are food webs we see now a result of past land-use or climate? I think the land-use analysis provides some insight on this but can the authors comment on this?
5. Overall, the figures are exceptional!
6. Figure 3 – these appear to be good models overall. I recommend the authors add a summary tables of the SEM results with estimates and confidence interval around them in appendix?

Shawn J. Leroux

Reviewer #2 (Remarks to the Author):

Review: Blue and green food webs respond differently to elevation and land use

This study is about the relationship of two contrasting types of food webs(terrestrial & aquatic) using structural properties across an environmental gradient, I think that the manuscript covers an important and interesting topic, which is needed to improve our knowledge of the responses of ecological networks to the many changes that anthropogenic activities are imposing over ecosystems.

My main concern is about the methods that in general are linear models and the structural properties seem to have a very strong nonlinear response to the main environmental variable that is elevation to many of the structural properties, this was presented in Figures 4 - 5 and S5 - S7, and noticed in the discussion (line 219). If this is true the assumptions of the SEM analysis and other linear analyses are probably not met. Another consequence of this nonlinearity is that at some range of elevation the correlations observed in SEM could be positive and in another range negative while the overall suggest a single type of correlation giving a misleading interpretation than the particular response of the structural properties is only one.

I suggest that the authors should show if the assumptions of the linear models are met (in the supplementary material), and if not take into account these nonlinearities, basically they shouldn't apply SEM, this seems to be more important at the global level, but could also be present at the land-use type level. They could also make a test of nonlinearity using GAMS, which can be implemented using R:

<https://stats.stackexchange.com/a/451170/45514>

Or more advanced nonlinear methods

<https://cran.r-project.org/web/packages/rEDM/vignettes/rEDM-tutorial.pdf>

See references [1,2]

The second concern is that is not clear if they analyze the absolute values of the Nestedness Modularity and Niche Overlap or the differences between the null models. They correctly made two null models, but there is an important difference in the Number of nodes and Connectance of the blue and green networks, as this has a major influence on other metrics connectance should be factored out from the analysis see [3] and references therein, primarily when they make comparison of slopes.

My recommendation for this manuscript is a major revision.

Minor comments

Line 288 "the warming effect on food webs can be moderated by human-caused" they are not moderated = mitigated by land-use because land-use have a strong effect.

Line 299 ARE vulnerable to anthropogenic...

Line 408 You also use Potential and realized generality that should be described in the main text methods, also the

Line 410 You should include generality in the PCA because it is an important metric.

line 441 realised and potential diet generality should be explained previously.

1. Sugihara, G., May, R., Ye, H., Hsieh, C., Deyle, E., Fogarty, M., et al. (2012). Detecting Causality in Complex Ecosystems. *Science* 338, 496–500. doi:10.1126/science.1227079.

2. Deyle, E. R., and Sugihara, G. (2011). Generalized Theorems for Nonlinear State Space Reconstruction. *PLOS ONE* 6, e18295. doi:10.1371/journal.pone.0018295.

3. Poisot, T., and Gravel, D. (2014). When is an ecological network complex? Connectance drives degree distribution and emerging network properties. *PeerJ* 2, e251. doi:10.7717/peerj.251.

Reviewer #3 (Remarks to the Author):

The authors have compiled an absolutely fantastic data set on the occurrence of species within terrestrial and aquatic organism groups across Switzerland. Based on trophic interactions inferred among 2016 plant, 191 butterfly, 109 grasshopper, 155 bird, 248 stream invertebrate, and 78 stream fish 346 species (the metaweb), they then examine how structural features of local realizations of the food web vary along environmental gradients. In doing so, they assume that any interaction ever documented (or inferred) to occur between two species anywhere will occur whenever they meet.

The paper builds on a series of (very) strong assumptions. In fact, there are NO direct observations of local interaction networks, only of species co-occurrence. All response variables are thus inferred from what will happen IF 1) the metaweb is correct and 2) links between species are deterministic, not probabilistic. Unfortunately, neither 1) nor 2) seem justified.

In terms of the metaweb built as the basis of all further analysis, the key point to note is that it builds on methods which cannot be replicated. The methods section suggest that the inference is defined in Suppl S1. Here, we find the statement that “The trophic interactions among focal groups were first assigned based on references, then further complemented from experts’ knowledge by the authors and collaborators.” Unfortunately, this means that none but the authors will know what was done and how. This is a frequent problem with older aquatic metawebs, but I am afraid it has lived its time. Unless the source of information can be retraced for each individual data point to AT THE VERY LEAST who felt it should be there, then this is not science but a personal opinion. This problem is far from not unique to the current study, and the same things has been done in multiple recent and high-profile papers. That does not make it any more legitimate.

To understand the extent of the problem, please note that for the aquatic food web alone, we have to make explicit decisions on the existence or not of associations between 248 stream invertebrate x 78 stream fish taxa = 20K links. For the terrestrial webs, we add a zero to that number. Who can possibly do that? Please note that all the metrics considered are exact representations on what specific links occur and what links are absent, so this is very far from hair splitting – it is the basis of the full analysis.

To understand some of the implications of variation in methods, resolution and data quality between taxa, please consider the impact of any assumptions – either implicit or explicit – as used by the experts in scoring the links as 0 or 1. These experts will likely vary between organism groups, and between the aquatic and terrestrial realm. Now, of the results, some general descriptors like “The nested and non-modular structure detected in the blue food webs (Fig. 5) well-echoes the reports in the literature [43, 44], reflecting that aquatic consumers are usually broad feeders with diets mostly constrained by body size (or say gape size) instead of other feeding traits [45, 46]. In contrast, the relatively high modularity observed in the green food webs (Fig. 5) suggests a relatively high proportion of specialised consumer-resource pairs within communities [47] (Fig. S7).” That is all very well – unless in scoring the links, the experts have used the assumption that aquatic consumers are gape limited in the first place. IF they have then

the results are solidly circular. And if they have NOT then HOW did they score the presence/absence of 20K links?

In terms of the notion that links between species be deterministic, not probabilistic, I cannot understand the reasoning. How could it possibly be that whenever two non-specialist species co-occur, they will be pulled together like magnets? As a thought, this strikes me as extremely unlikely. And how could it be that while the environment comes with major effects on the set of species (co-)occurring, it has no impact on their interactions? These ideas have been explored and challenged by e.g. Gravel et al. 2019. *Ecography*, doi.org/10.1111/ecog.04006 and Cirtwill et al. 2019. *Methods in Ecology and Evolution*, doi: 10.1111/2041-210X.13180. On lines 312-319, the authors advance some caveats related to this, but do nothing to dispel them.

In conclusion, my main concern then is that given the scarcity (i.e. lack of) of direct data on the focal metrics analysed, the overall study will mainly reflect its main assumptions – rather than revealing anything new of nature. Where the patterns detected feel robust, that is basically because the associations built on are logical outcomes of associations so well established from before that there is little to write home about anyway.

As a (much) secondary concern, the introduction will need substantial editing by an attentive copy editor – or much rather by an attentive author. The first sentence of the main text hits the reader like the brick to the head, and it does not get any easier from there. Most sentences have 3-5 commas, and when commas do not suffice the authors have introduced long dashes just to avoid ever breaking the sentence. It is almost like the authors worried that if they express themselves in simple words and clear sentences, then the science would turn simplistic, too? Here, the methods and results sections offers a clear respite: they are clear, succinct and well structured. I will assume that different people wrote different parts, and ask the person writing the methods and results to take equal responsibility for the other sections.

Responses to comments

We thank the Editor and Reviewers for their constructive and supportive comments, which we all integrated. We also resolved aspects that were based on possible misunderstandings and a more general critique to the metaweb approach chosen. As this goes beyond the focus of our manuscript, we describe how the method chosen is appropriate in our case, yet we do not think that a fundamental discussion on its suitability should be the focus of our work.

The comments received, and our subsequent additional analyses, e.g., on the combined blue-green food webs, and the text added/modified have strengthened our manuscript. For these, we would like to thank the Reviewers in particular.

We here reply to the Editor's and the Reviewers' comments on a point-by-point basis. The comments are in **black**, our responses in **blue**. All line numbers, if mentioned in the responses, refer to the new line numbers after manuscript revision, unless specifically noted otherwise. All corresponding revisions are highlighted with **yellow background** in the manuscript.

Comments from the Editor

1. You will see that, while the reviewers find your work and dataset of interest, they raise substantive concerns that cast doubt on the strength of the novel conclusions that can be drawn at this stage. Unfortunately, these reservations, particularly those from reviewer #3, are sufficiently important to preclude publication of this study in Nature Communications.

Response: We thank the Editor for their efforts for handling our manuscript. We strongly regret learning this decision of rejection, especially as reviewers #1 and #2 were very positive and made very constructive and implementable comments.

The critical comments raised by Reviewer #3 (next to some constructive and implementable comments) emerged from possible misunderstandings of our actual research methods, and tended to be a general disagreement of the metaweb approach chosen (thus being a generic critique based on personal opinion, beyond our actual work), even though this approach represents an emerging research direction. We had already highlighted the main assumptions of the metaweb approach in the previous manuscript version, yet have now expanded this part to ensure that all assumptions are well-presented and clear. Importantly, this approach becomes widely used in recent ecological analyses (e.g., Albouy et al. 2019, Neff et al. 2021, Bauer et al. 2022, Galiana et al. 2022). For large-scale analyses with reliable co-occurrence information as ours, there is, until proven differently, no better alternative approach for now.

We are confident that our arguments in the following responses justify and corroborate the methodology chosen and scientific novelty of our research. We are also confident that our responses here will help future review process with our new submission to avoid the same conceptual pitfalls.

Albouy, Camille, et al. "The marine fish food web is globally connected." *Nature Ecology & Evolution* 3.8 (2019): 1153-1161.

Neff, Felix, et al. "Changes in plant-herbivore network structure and robustness along land-use intensity gradients in grasslands and forests." *Science Advances* 7.20 (2021): eabf3985.

Bauer, Barbara, et al. "Biotic filtering by species' interactions constrains food-web variability across spatial and abiotic gradients." *Ecology Letters* (2022).

Galiana, Núria, et al. "Ecological network complexity scales with area." *Nature Ecology & Evolution* (2022): 1-8.

Comments from Reviewer #1 (Prof Shawn J. Leroux)

Summary -- Biodiversity is under threat from global changes and there is a pressing need to predict how community will respond to these changes. A key part of biological communities is the interactions that bind species yet we have only recently begun to study how these interactions are changing. The few studies that investigate this problem focus on bipartite webs in one ecosystem. In this contribution, the authors assemble an outstanding data set on food webs in terrestrial (green) and aquatic (blue) ecosystems to investigate how elevation, climate and land-use impact food web properties. The authors offer a tantalizing set of results that demonstrate key drivers of food web variation both within ecosystem (i.e., across blue webs or across green webs) and across ecosystems (i.e., blue vs green webs).

Response: We thank you for your very positive, supportive and constructive review, which was much appreciated. We have integrated all your comments, and added the additional analyses suggested. This strengthened our results and conclusions, which we are very thankful for.

1. This is an impressive study! The major strengths are the strong data sets, robust analysis of both within and across ecosystem drivers of food web change, and insightful interpretations that will make this study a foundation for future work. The authors have identified a clear gap and developed a strong study to fill this gap.

Response: We thank for the Reviewer's very positive appraisal, and the identification of the studies significance. We are happy to hear that you see this being a foundational study for future work.

2. At first, I thought the authors were going to present full connected green and blue webs – so I was a bit disappointed that this was not done. That said, I fully value the approach taken and see it as a logical first step. But I want to know more about the potential connections across green-blue webs. Are there really few sampling sites that accommodated these potential connections. In this system, is it possible that the species that connect green and blue may be critical species to consider for landscape management?

Response: Indeed, this would have been a very interesting approach. Yet, this would have required the co-sampling of blue- and green food webs in the same spatial locations, which unfortunately was not the case with our occurrence datasets.

That said, we agree that this blue-green connected (or say merged) perspective could be discussed within our blue vs. green scope. We therefore carried out additional analyses with our data at a rougher 5x5 km² spatial resolution (which allowed 94 grids of blue and green webs to be co-analysed, though all this information become less representative compared to our main analyses). We now discussed this part in the main text (L312 onwards), and provided results and further-detailed discussion in the SI at Section S4. As you will see, these additional analyses are in line with our main findings, yet they have much less power due to the lower number of sited that could be co-analysed. We do, however, agree that this is an important and interesting addition which we happily added.

3. The data are super impressive – wow. But clearly these data are only a subset of the webs – particularly in terrestrial systems. Do we know anything about how higher level species (ie predators) are distributed across the terrestrial gradients? I was surprised not to see any discussion of this.

Response: This is a good and valid point. There are more species (and more taxonomic groups) than the ones covered (e.g., microbes are completely missing, as are several insect groups). We highlight that the total 2797 species (taxa) covered is indeed an unprecedented resolution, yet does not cover all taxa yet. We simply had to rely on the organismal groups for which the national database (BDM and

info fauna) had accurate occurrence information. We also had to exclude some groups whose occurrences were available, but we lacked detailed information on trophic interactions matching such occurrences to integrate them into the current study (e.g., some terrestrial tetrapods, even with the TETRA-EU metaweb (Maiorano et al, 2020) the trophic resolution would be too rough). We thus consider our compromise between taxa coverage vs. resolution/reliance of the trophic information was proper. We thank the Reviewer for raising this point, and we have now incorporated relevant discussion to the revised manuscript (L298 onwards).

Maiorano, L., Montemaggiore, A., Ficetola, G. F., O'connor, L., & Thuiller, W. (2020). TETRA-EU 1.0: a species-level trophic metaweb of European tetrapods. *Global Ecology and Biogeography*, 29(9), 1452-1457.

- a. The environment and species are changing. I would like to know how the timing of sampling of the different biotic and abiotic data compare here. There is some mention of the timing of some components of the abiotic side but the reader needs to dig into the references to get more information on most other pieces of data. I recommend the authors consider adding a table with sampling period in appendix. If sampling of biotic occurs in a very different time frame than climate data, then the authors might consider constraining their analysis to data that are sampled in the same period. Otherwise, I wonder if variation in sampling period might be adding some noise to these relationships.

Response: We thank the Reviewer for this suggestion. We now added a supplementary table (S1) to provide such information. Taxa occurrence from different sources were aggregated over non-identical time spans but generally within/encompassing 2000–2020. All environmental measures were averaged over 2005–2015, which was chosen to be as align as possible with the biotic information. Considering the large spatial scale we were looking at, we do not expect that the detected patterns would have strong systematic temporal variations over such a time window.

- b. Do the authors have replicates in the sampling of blue webs – ie two sampling sites on one stream? If so, did the authors treat this nested structure in their analyses?

Response: The aquatic taxa occurrences were on a 1×1 km² grid basis without linking to the identities of sampled streams. From the grids' locations, as shown in Figure 2, these sample sites were at least several kilometres away from each other (Euclidian between-site distance ~2.8 km in three pairs, all else were greater). While few sites belong to the same larger catchment systems, they were never from the same smaller sub-tributary systems. In any case, the potential autocorrelation between sites due to being connected by the same flowing water should have been diluted across the large spatial scale we are looking at. We therefore did not dig into the stream identities of each data points to consider a nested data structure, but just treat them as independent from each other.

- c. lines 348-350. How does making this assumption influence your results – this sets a strict number of nodes as basal resources so will constrain your metrics. It is still a small number relative to the total number of nodes but might be worth a discussion (in appendix?). I would expect higher connectance and lower modularity in blue vs green if basal resources are unchanging in blue. And I believe this is what you found. This would not affect intraspecific (ie blue vs blue OR green vs green) web comparisons but maybe interspecific (ie blue vs green).

Response: We agree with the Reviewer's suggestion and have now added relevant discussion in the main text (L298 onwards). For the detected aquatic primary consumers, they certainly were feeding on some basal resources locally, however we had no data on the occurrence/composition of those actual basal resources. Thus, aggregating all possible basal

resources into fixed mega nodes was a needed simplification to keep the detected aquatic primary consumers in the local food webs (otherwise they would falsely become isolated consumers without resource). Indeed, as the Reviewer also recognised, this would not affect intra-biome but inter-biome comparisons. This was exactly why throughout this study we emphasised the qualitative pattern-wise differences (e.g., how a metric changed along a gradient) but not the quantitative metric-wise readings' differences between the blue and green.

- d. lines 394-408, Figure 3B. I recommend the authors walk the reader through Fig 3B in appendix. I think this will help make the work more accessible to a broad audience. There is a lot under the hood here! The authors are use common food web metrics. Did the authors have any a priori predictions for relationships or was this purely exploratory?

Response: We thank for the suggestion and have added relevant texts in the main text *Methods* (L459 onwards). Regarding Figure 3B, we believe that the dependencies from elevation (an environmental predictor) to food-web metrics (ecological responses) are intuitive. The dependencies among food-web metrics were assigned with the principle of pointing from relative lower-level properties to higher-level ones. That is, from number of nodes (purely determined by nodes) to connectance (determined by numbers of nodes and links), further to nestedness and modularity (holistic topologies, determined further by the arrangement of links), then to niche overlap (ecological functional outcome). Such a dependency structure was referred from literature as cited in the manuscript (refs [12, 65]).

- e. lines 434-444 This is a nice explanation of the value of the two randomized webs here.

Response: We thank for this positive comment.

4. The discussion is very well done – it blends specific results with broad interpretations (ie beyond their system). The land-use type analysis is quite fascinating and rare at this scale. But I am wondering about history or legacy effects here. To what extent are food webs we see now a result of past land-use or climate? I think the land-use analysis provides some insight on this but can the authors comment on this?

Response: This is a very interesting comment. Actually, it may be a more general question in ecology overall: how much do contemporary community patterns reflect current vs. past imprints by environmental drivers. We unfortunately do not have the temporal resolution with respect to the species turnover, and thus this simply cannot be addressed by our datasets. We now added a sentence (L294) stating that contemporary species occurrences/community structures (and thus eventually also food-web structures) are both a result of contemporary environmental and biotic drivers, as well as of past drivers, leaving a legacy.

5. Overall, the figures are exceptional!

Response: We thank for this positive comment. We spent a lot of time working on these figures and are thus happy to hear that you find them appealing.

6. Figure 3 – these appear to be good models overall. I recommend the authors add a summary tables of the SEM results with estimates and confidence interval around them in appendix?

Response: Revised accordingly (Tables S7–S12).

Comments from Reviewer #2

This study is about the relationship of two contrasting types of food webs(terrestrial & aquatic) using structural properties across an environmental gradient, I think that the manuscript covers an important and interesting topic, which is needed to improve our knowledge of the responses of ecological networks to the many changes that anthropogenic activities are imposing over ecosystems.

Response: We thank the reviewer for their positive and constructive evaluation of our study.

1. My main concern is about the methods that in general are linear models and the structural properties seem to have a very strong nonlinear response to the main environmental variable that is elevation to many of the structural properties, this was presented in Figures 4 - 5 and S5 - S7, and noticed in the discussion (line 219). If this is true the assumptions of the SEM analysis and other linear analyses are probably not met. Another consequence of this nonlinearity is that at some range of elevation the correlations observed in SEM could be positive and in another range negative while the overall suggest a single type of correlation giving a misleading interpretation than the particular response of the structural properties is only one.

I suggest that the authors should show if the assumptions of the linear models are met (in the supplementary material), and if not take into account these nonlinearities, basically they shouldn't apply SEM, this seems to be more important at the global level, but could also be present at the land-use type level. They could also make a test of nonlinearity using GAMS, which can be implemented using R (<https://stats.stackexchange.com/a/451170/45514>), or more advanced nonlinear methods (<https://cran.r-project.org/web/packages/rEDM/vignettes/rEDM-tutorial.pdf>). See references [1,2].

[1]. Sugihara, G., May, R., Ye, H., Hsieh, C., Deyle, E., Fogarty, M., et al. (2012). Detecting Causality in Complex Ecosystems. *Science* 338, 496–500. doi:10.1126/science.1227079.

[2]. Deyle, E. R., and Sugihara, G. (2011). Generalized Theorems for Nonlinear State Space Reconstruction. *PLOS ONE* 6, e18295. doi:10.1371/journal.pone.0018295.

Response: We thank for the constructive comments. Indeed, across the whole elevational range, food webs tended to exhibit nonlinear responses. We therefore already analysed them with GAMs (Figures 4–5) and, alongside the overall SEM, carried out SEM specific to each land-use type (Figure S1) to indicate that the detected relationships in the overall SEM were not in conflict with those detected at the land-use type level (i.e., elevational sections). Nonetheless, we agree that model assumptions should be tested. The piecewise SEM allows dependencies to be derived by LMMs/GLMMs, we thus reran the overall SEM analysis taking land-use type as a random effect, and updated the results accordingly. The corresponding linear model assumptions were largely met (Figures S12–17), and our findings remain qualitatively the same. For linear model analyses on food-web metric against elevation (Figures S3–S4), we tested the linear assumption using GAMs as the Reviewer suggested. While nonlinearity was detected in individual cases (mostly weak), the majority was linear (Tables S5–S6). Thus, we think using linear models throughout was a valid approach based on the principle of parsimony, especially given that our interpretations/conclusion were not drawn purely from one set of comparisons.

2. The second concern is that is not clear if they analyze the absolute values of the Nestedness Modularity and Niche Overlap or the differences between the null models. They correctly made two null models, but there is an important difference in the Number of nodes and Connectance of the blue and green networks, as this has a major influence on other metrics connectance should

be factored out from the analysis see [3] and references therein, primarily when they make comparison of slopes.

[3]. Poisot, T., and Gravel, D. (2014). When is an ecological network complex? Connectance drives degree distribution and emerging network properties. *PeerJ* 2, e251. doi:10.7717/peerj.251.

Response: We fully agree that the number of nodes and connectance can influence other food-web metrics as emerging properties. We therefore already considered (and revealed) such influences with the SEMs (Figures 3 & S1), and when looking at elevational patterns of individual metrics, we made each of the generated null webs have the same (i.e., fixed) number of nodes and connectance with its empirical counterpart (Figure 5). In other words (as elaborated at L483 onwards), the influences of the changing number of nodes and connectance along elevation on any web metric would have been captured by the null webs' elevational patterns.

We agree that there are alternative ways to code our observations (e.g., converting the observed values to Z-scores against null values to standardise/factor out the node and connectance influences). However, with the aim of revealing empirical community patterns, we decided to report the observed webs alongside the null ones. This way allowed ecologically more-sensible understandings such as "food webs become more nested because they shrink and lose generalists" (e.g., around L175), instead of correct yet ecologically enigmatic ones such as "all else being equal, food web nestedness will not change".

3. Line 288 "the warming effect on food webs can be moderated by human-caused" they are not moderated = mitigated by land-use because land-use have a strong effect.

Response: Thank you. We have rephrased to clarify that we were not referring to mitigation, but that the influences of environmental factors on the food webs were depending on land-use types (L293).

4. Line 299 ARE vulnerable to anthropogenic...

Response: Revised accordingly (L325).

5. Line 408 You also use potential and realized generality that should be described in the main text methods, also the Line 410 You should include generality in the PCA because it is an important metric. Line 441 realised and potential diet generality should be explained previously.

Response: The generality analyses were conducted on a taxa-group basis, because we considered it was more ecologically meaningful this way than having an overall mean value as the representative of a local food web with multiple taxa groups. Thus, generality did not fit to the PCA with other holistic metrics derived on a whole-web basis, and with its plenty of figures, it suited best to the supplementary information. We explained relevant information when the terms first introduced in the main text *Methods*.

Comments from Reviewer #3

The authors have compiled an absolutely fantastic data set on the occurrence of species within terrestrial and aquatic organism groups across Switzerland. Based on trophic interactions inferred among 2016 plant, 191 butterfly, 109 grasshopper, 155 bird, 248 stream invertebrate, and 78 stream fish 346 species (the metaweb), they then examine how structural features of local realizations of the food web vary along environmental gradients. In doing so, they assume that any interaction ever documented (or inferred) to occur between two species anywhere will occur whenever they meet.

Response: Thank you. We appreciate your enthusiastic feedback, and agree that this is a unique and fantastic dataset that allows an unprecedented metaweb analysis across hundreds of sites, thousands of species, and covering both aquatic and terrestrial systems. The metaweb approach we adopted is a clear development from well-established research in community ecology of using species functional traits to understand biotic interactions and community assembly (e.g., Bartomeus et al. 2016, Godoy et al. 2018), while it makes synthesis directly of potential interactions instead of traits. Please find our following responses where we provide more details of this approach and supporting references.

We are thankful for your constructive feedback and have addressed it all, meanwhile we have clearly justified our study approach to resolve your concerns.

Bartomeus, Ignasi, et al. "A common framework for identifying linkage rules across different types of interactions." *Functional Ecology* 30.12 (2016): 1894-1903.

Godoy, Oscar, et al. "Towards the integration of niche and network theories." *Trends in Ecology & Evolution* 33.4 (2018): 287-300.

1. The paper builds on a series of (very) strong assumptions. In fact, there are NO direct observations of local interaction networks, only of species co-occurrence. All response variables are thus inferred from what will happen IF 1) the metaweb is correct and 2) links between species are deterministic, not probabilistic. Unfortunately, neither 1) nor 2) seem justified.

Response: We respectfully disagree. Our (most parsimonious) assumption is that interactions may only occur when species co-occur. Importantly, the interactions and co-occurrences we used have both been observed. The vast majority of the interactions is from Switzerland or at least Central Europe, while the co-occurrences from Switzerland. At the spatial range we focus on, these information are based on strong and broad empirical supports and a wide range of natural history observations, and therefore are plausible and realistic. We do not assign interactions unrealistically to species that never co-occur, nor based on probabilistic approaches (e.g., subsampling, inferential trait-matching) without *a priori* mechanistic expectations.

Equally importantly, we do not claim all interactions must occur. We realise there is maybe only a subset of them realised, but at least the empirically rooted interactions and co-occurrences together give the (realistic) boundary conditions for the potential interaction realisation. These assumptions are broadly used in metaweb approaches (e.g., Bauer et al. 2022, Galiana et al. 2022, alongside others already cited in the manuscript), and we have clearly stated them in the manuscript (L349 onwards) along with corresponding limitations. We will justify these assumptions in further details below when the Reviewer expand the comments.

As a final argument to this comment: if the strong empirical support of our metaweb does not make it believable, probably none of the existing empirical food web dataset would be considered as correct based on the Reviewer's comment. Even for local webs that based on direct observations, these observations could cover only a subset (likely a small part) of the occurring species and their interactions, thus the webs are surely undersampled (e.g., Goldwasser & Roughgarden 1997, Wood et

al. 2015), and the interactions then extrapolated to the individuals of these species at a site. To us, what matters is how to address ecological questions, validly, with these known-biased understandings, in which respect we believed our approach is a proper choice.

Bauer, Barbara, et al. "Biotic filtering by species' interactions constrains food-web variability across spatial and abiotic gradients." *Ecology Letters* (2022).

Galiana, N ria, et al. "Ecological network complexity scales with area." *Nature Ecology & Evolution* (2022): 1-8.

Goldwasser, Lloyd, and Jonathan Roughgarden. "Sampling effects and the estimation of food-web properties." *Ecology* 78.1 (1997): 41-54.

Wood, Spencer A., et al. "Effects of spatial scale of sampling on food web structure." *Ecology and Evolution* 5.17 (2015): 3769-3782.

2. In terms of the metaweb built as the basis of all further analysis, the key point to note is that it builds on methods which cannot be replicated. The methods section suggest that the inference is defined in Suppl S1. Here, we find the statement that "The trophic interactions among focal groups were first assigned based on references, then further complemented from experts' knowledge by the authors and collaborators." Unfortunately, this means that none but the authors will know what was done and how. This is a frequent problem with older aquatic metawebs, but I am afraid it has lived its time. Unless the source of information can be retraced for each individual data point to AT THE VERY LEAST who felt it should be there, then this is not science but a personal opinion. This problem is far from not unique to the current study, and the same things has been done in multiple recent and high-profile papers. That does not make it any more legitimate.

Response: We respectfully disagree that our methods cannot be replicated. Firstly, all data used are clearly reported, trackable, and accessible. The "references" mentioned in the quoted sentence, from which we established the majority of our metaweb trophic links among the focal taxa groups, were already clearly given in Table S1. Also, we provided our complete metaweb in Section S1 (at the online repository). Readers of this manuscript can easily access the trophic-link information therein, reproduce all analyses (with the attached codes), or apply the metaweb to do further research. The condition was therefore far from the Reviewer's description "none but the authors will know what was done and how", which we respectfully disagree.

For trophic links that were not covered by the listed references, and thus were assigned by the authors' and collaborators' expert knowledge, we are open for being contacted for any relevant concern or issue. To facilitate that, also in respond to the Reviewer's concern, we have now added a column to our metaweb entries to indicate whether the link was assigned by the authors.

3. To understand the extent of the problem, please note that for the aquatic food web alone, we have to make explicit decisions on the existence or not of associations between 248 stream invertebrate x 78 stream fish taxa = 20K links. For the terrestrial webs, we add a zero to that number. Who can possibly do that? Please note that all the metrics considered are exact representations on what specific links occur and what links are absent, so this is very far from hair splitting – it is the basis of the full analysis.

Response: We agree with the Reviewer that food-web metrics consider both the presence and absence of the links, but with regard to the Reviewer's following comments, especially "For the terrestrial webs, we add a zero to that number. Who can possibly do that?" we think that there was a misunderstanding of our approach, which we now have clarified. What we analysed were not the metaweb but the occurrence-informed local food webs, blue and green separately. In other words, we did not assign zeros to potentially 20k aquatic links (i.e., all links in the aquatic metaweb) to the green local food webs then calculated some zero-inflated metrics, because those aquatic species simply did not occur nor even involve in the local green communities. With a matrix expression of food web (i.e., diet matrix), each our local food web would be a matrix with column and rows being the species that were

empirically recorded occurring at the focal site, while those did not occur (or even from another biome) would not present, nor do their potential interactions. We believe the Reviewer also recognise that such presence-observation based diet matrices are the most common food-web coding used in almost all food-web studies. This information was already clearly explained in our *Methods* section (which receives the Reviewer's appraisal in the last comment) and further illustrated in Figure 1.

4. To understand some of the implications of variation in methods, resolution and data quality between taxa, please consider the impact of any assumptions – either implicit or explicit – as used by the experts in scoring the links as 0 or 1. These experts will likely vary between organism groups, and between the aquatic and terrestrial realm. Now, of the results, some general descriptors like “The nested and non-modular structure detected in the blue food webs (Fig. 5) well-echoes the reports in the literature [43, 44], reflecting that aquatic consumers are usually broad feeders with diets mostly constrained by body size (or say gape size) instead of other feeding traits [45, 46]. In contrast, the relatively high modularity observed in the green food webs (Fig. 5) suggests a relatively high proportion of specialised consumer-resource pairs within communities [47] (Fig. S7).” That is all very well – unless in scoring the links, the experts have used the assumption that aquatic consumers are gape limited in the first place. IF they have then the results are solidly circular. And if they have NOT then HOW did they score the presence/absence of 20K links?

Response: Our previous response has presumably clarified the possible misunderstanding that led to the question of “And if they have NOT then HOW did they score the presence/absence of 20K links?”

For the rest of this comment, we would like to respectfully stand up for all the experts with extensive naturalist knowledge (some of us in the author team belong to those, and have assembled many of the data points): These experts have collected these interaction data, and it is not derived from a circular approach. The comment implies that we assigned trophic links based on certain assumptions (e.g., gape size), which is actually not true. As in our response to the Reviewer's comment 2, the majority of our trophic relationships among the focal taxa were obtained from reliable literature based on a large quantity of empirical observations (i.e., first-hand observations of these interactions). Obviously, for such a broad range and high number of taxa, there must be different sources of this information, as they require very different taxonomic and identification expertise. Thus, bringing together this primary literature for the wide range of taxa, and complement it with our own expertise and observations, is actually the best one can do and a major step forward.

For links beyond the literature and assigned by experts (authors and collaborators), these were still based on empirical observations and/or unpublished data accumulated over considerable field research experience, instead of just simple trait considerations. Though we felt unnecessary, it could be revealed that among the authors and collaborators, JB and OS are leading fish experts, MG is a leading terrestrial herbivore expert, SK is an aquatic invertebrate expert, FA is a leading aquatic invertebrate and Lepidoptera expert, and R. Ayé is a leading bird expert. Thus, we as authors actually know the focus species very well. We would not decide to take a metaweb approach if we were not such a diverse collaborating group that can cover all needed trophic understandings. We would also like to highlight that the vast majority of knowledge in ecology and biodiversity sciences is coming from experts, whose broad knowledge is essential for the understanding of these systems (and also corroborating our interpretation), and we feel that it is very important and valuable to integrate this knowledge.

As it may have been based on misunderstandings, we thank the Reviewer for the potential doubt about the experts' knowledge part, and we have revised to strengthen relevant statements in the manuscript for clarification (L339 onwards).

5. In terms of the notion that links between species be deterministic, not probabilistic, I cannot understand the reasoning. How could it possibly be that whenever two non-specialist species co-occur, they will be pulled together like magnets? As a thought, this strikes me as extremely unlikely. And how could it be that while the environment comes with major effects on the set of species (co-)occurring, it has no impact on their interactions? These ideas have been explored and challenged by e.g. Gravel et al. 2019. *Ecography*, doi.org/10.1111/ecog.04006 and Cirtwill et al. 2019. *Methods in Ecology and Evolution*, doi: 10.1111/2041-210X.13180. On lines 312-319, the authors advance some caveats related to this, but do nothing to dispel them.

Response: As the Reviewer noted, we had revealed the assumptions of the metaweb approach in our manuscript (L349 onwards), and interpreted our results accordingly, so the readers are fully informed about the conditions under which we detected the patterns as well as their reasonable ecological interpretations. However, to avoid that this is overseen, we expanded on the description of assumption.

Echoing our response to the Reviewer's comment 2, we did not imply that all the interactions of co-occurring species "must" occur. Rather, there is the potential of them to occur, bounded by that the interactions *per se* are known eligible (i.e., once empirically observed) and the species involved co-occur (i.e., empirically monitored). All of our analyses on the potential interactions realised at a local site based on co-occurrence were therefore plausible and, importantly, coherent and non-biased across all sites. Also, all interpretations did not hinge on the interactions actually occurring all the time. We of course acknowledge that environment may influence the realised species interactions (though not necessarily greatly, see Bauer et al., 2022). However, as our metaweb trophic interactions were based on empirical observations, we do not see making the co-occurring consumer-resource pair interact was an "(quoted) extremely unlikely" assumption. Across the spatial and temporal scale of each local grid, collapsing all intra-specific variations (including trophic-determining traits, life stages, etc.) of local populations therein, just one pair of (plausible consumer-resource) individuals that meet under the correct conditions (e.g., with a suitable body-size ratio) to perform the once-observed trophic interaction would already authorise such a link to be realised in our presence-based local diet matrix. Our trophic relationship assignment is therefore not only comparatively likely, but also more conservative than other approaches based on just one or two selected trait-matchings.

We fully follow the reference studies suggested by the Reviewer (even already cited one in our original manuscript). However, we disagree that taking probabilistic approaches without *a priori* mechanistic understandings would better answer our research question, particularly given the high quality of our empirical data.

Whether to use probabilistic approaches can be boiled down to whether occurrences and interactions are "true and known". With an extremely strict perspective, none of our understanding of natural food webs, even empirically measured, could ever be perfectly "true and known", as no one can sample all the organisms and keep track of their real life-long diets (thus Cirtwill et al. modelled probabilities of detection of these information). Clearly, this does not mean that we should fully abandon empirically observed information. Instead, this implies that probabilistic approaches can be appropriate complements to empirical data when the empirical data themselves are too rare or too uncertain to answer the focal question. To the occurrences end, probabilistic SDM could be a way to achieve the desired spatial coverage at places without empirical occurrence understanding. However, our focal ~900 sites were systematically monitored, having among the best-quality species occurrences at such a spatial scale. Our occurrence understanding was thus surely close to "true and known". To the interactions end, as above stated, our metaweb was based on previously recorded interactions (also recognised as a valid approach in Gravel et al., 2019), giving much higher reliability than purely trait- or phylogeny-inferred ones (or their corresponding probabilistically derived ones). These interactions

are thus surely “known” and arguably more “true” than inferred otherwise. Together, our approach combined the closest approximates toward the “true and known” in both occurrences and interactions ends. We were not aiming at re-filling missing links from undersampling in localised food webs (see Cirtwill et al.’s model), nor constructing food webs with significant uncertainties in the quantification of both occurrences and interactions (see Gravel et al.’s model). The scale of our research question, and the strong empirical basis of our data, make our deterministic metaweb approach appropriate. There was no obvious necessity to add probabilistic layers to either our occurrences or interactions at the costs of masking precious empirical information. Finally, even purely from a methodological point of view, taking probabilistic approaches would require additional assumptions on the specifics of relevant subsamplings. Those would be assumptions without direct justification and thus not warranted from a parsimony perspective.

Bauer, Barbara, et al. "Biotic filtering by species' interactions constrains food-web variability across spatial and abiotic gradients." *Ecology Letters* (2022).

6. In conclusion, my main concern then is that given the scarcity (i.e. lack of) of direct data on the focal metrics analysed, the overall study will mainly reflect its main assumptions – rather than revealing anything new of nature. Where the patterns detected feel robust, that is basically because the associations built on are logical outcomes of associations so well established from before that there is little to write home about anyway.

Response: With our responses to the above comments 1–5, we should have clarified the possible misunderstandings: actually, all our data used were direct data based on observed occurrence and interactions (just that not all were observed by us). We were very aware of what can be done with our data and methods, as well as the relevant limitations, thus we expanded the manuscript carefully and our statements were nowhere close to being over-interpreted. We do not see “the scarcity (i.e. lack of) of direct data on the focal metrics analysed” while looking at the occurrence and trophic interaction information we compiled. The Reviewer would for sure agree that to empirically sample >900 individual local food webs across Switzerland would be an impossible mission. We then argue that our approach, as empirically rooted as the data could be, is the best next (and importantly, feasible) approach to address this research question at such a large spatial scale. Similarly, with our above responses, we definitely dispute the criticism of “mainly reflect its main assumptions – rather than revealing anything new of nature”. The blue and green food-web patterns revealed by our study is clearly an important new finding to the field, and we have above clearly clarified why these patterns detected by the metaweb approach were not assumption-driven. We regret seeing such an arbitrary conclusion, presumably based on accumulated false assumptions and possible misunderstandings regarding our data and the actual study approach.

7. As a (much) secondary concern, the introduction will need substantial editing by an attentive copy editor – or much rather by an attentive author. The first sentence of the main text hits the reader like the brick to the head, and it does not get any easier from there. Most sentences have 3-5 commas, and when commas do not suffice the authors have introduced long dashes just to avoid ever breaking the sentence. It is almost like the authors worried that if they express themselves in simply words and clear sentences, then the science would turn simplistic, too? Here, the methods and results sections offers a clear respite: they are clear, succinct and well structured. I will assume that different people wrote different parts, and ask the person writing the methods and results to take equal responsibility for the other sections.

Response: Thank you. We have carefully revisited the text, and ensured it is coherent and correct from a linguistic perspective. Actually, it was the same person, the first author, who drafted the whole manuscript with inputs from co-authors, but we now paid extra attention to make the introduction more readable.

Reviewer #1 (Remarks to the Author):

The authors have completed a comprehensive revision that addresses all concerns raised by the three reviewers. I appreciated the additional details on methods and justification for data and analysis choices. In addition, the new analyses on connections between green and blue webs and non-linear relationships was well done– thank you.

The authors also provide a useful rebuttal to the concerns of Reviewer #3. I fully agree with the authors on all points. Food webs are a known unknown - we can never know all possible interactions and this is why we have developed many ways to quantify potential interactions. Even if a group was capable of empirically sampling nearly 1000 sites (for many years each!), they would still only capture a subset of all possible interactions. Yet, we are inherently interested in how food webs change across space and time. The authors use a metaweb to define “potential” interactions and they are explicit about the limitations of this approach. To me, this is a robust use of a metaweb and the clear communication of its use is a real strength of the paper – the authors identify the limitations and interpret their findings in light of this. To me, this is more robust than authors considering their empirically intensive sampling to represent “truth”. Most importantly, if a reader disagrees with the metaweb, the authors have provided all the data and references to support the links in their web. Their Rdata files are crystal clear and make this work reproducible. It seems Reviewer #3 may simply be against food web studies or spatial food web studies as they have not provided any concrete suggestions to address the various issues they identify. A probabilistic approach is not the solution here as the authors have clearly argued.

Reviewer #2 (Remarks to the Author):

I congratulate the authors on doing a thorough job of addressing the points I raised in the previous review round and making a solid defence of some of their methods. I think that the manuscript is generally improved. I would like to repeat that the topic is important and interesting, and will enhance our knowledge about food webs changes over natural and human induced environmental ranges. Thus the ms will be of interest to a broad readership, and it's worth being published in Nature Communications.

Leonardo A. Saravia

Reviewer #3 (Remarks to the Author):

The authors have now revised their interesting manuscript, and provided a strong rebuttal of the main criticism advanced. As I am Reviewer #3, I note that the main rebuttal of my own critical comments frames them as being based on a misunderstanding – which I do not think they were.

Our main source of disagreement can be simplified into two key points:

First, I claim that link structure, and thereby food web structure, cannot be inferred from species co-occurrence alone (which is the key assumption behind the “metaweb approach”): two species occurring the same place is simply insufficient proof that they interact. And if we lack such proof, then the actual data is on species pools (which can be conveniently analysed) and not on network structure (as the authors claim and model).

Second, I claim that a metaweb based on directly observed and published evidence of who interacts with whom cannot be replicated if and since it has been mixed with an unknown and undocumented proportion of expert opinion.

In response, the authors frame my criticism as being “a generic critique based on personal opinion, beyond [their] actual work, even though this approach represents an emerging research direction”. I respectfully disagree with them as respectfully as they disagree with me, and – most

importantly – see no more trace of a personal opinion in the critique advanced than in the responses proffered.

The first main defence of the current approach is that “other people do the same” – with which I agree. (In fact, I will volunteer a key reference from last week to add to the author’s list of other people who did the same, i.e. Bauer et al. 2022 Ecology Letters <https://onlinelibrary.wiley.com/doi/full/10.1111/ele.13995> . The crux here, though, is that pointing to some precedent does not validate an approach, since other people can make errors, too. Just look at what GBIF data have been used for by happy number crunchers just “because they are there”... What is more, the validity of an approach will vary with what it is used FOR.

The second main defence is that “this is the best we can do since surely, we cannot realistically collect direct evidence on the phenomenon that we claim to be analysing” [which is food web structure]. I accept that this is the case, but if we cannot realistically collect direct observations of something, then that does not in itself justify the use of other data to approximate it.

The third main defence is that the reconstruction of the metaweb can be repeated based on the description provided. I do not see how, since links based on expert opinion cannot be separated from links based on direct evidence? In support of the validity of their data, the authors advance the classical argument of “trust us, we are experts” – which I fully believe, but which is still off the point.

The fourth main defence is that the assumptions are clearly spelled out, and that it is then up to the reader to judge. With this I partly agree: the assumptions are clearly spelled out at the outset of the paper. My claim is still that they are unlikely to be true, and that this undermines the conclusions – which are not framed as being conditional on strong assumptions.

The authors’ claim is then that 1) these are assumptions which simply have to be made for us be able to approach the focal topics with the data at hand [with which I fully agree], and 2) that if they are true then blue and green food webs exhibit all sorts of structural and ecological properties along elevation and among various land-use types (which is very cool if true).

What I believe is that the data suffice for analysing environmental imprints on blue versus green species pools, whereas the authors believe that they suffice for analysing network structure. Since the real data needed to show who is right are critically missing (which is indeed my whole point), I guess the argument boils down to i) whether or not the main assumptions are stated clearly enough, ii) whether or not the key findings are consistently framed as being conditional on these assumptions, all the way to the concluding section (on which I disagree), iii)) whether or not the key assumption are likely to hold true (which I doubt), and iv) whether or not the results are robust to deviations from the main assumptions (which I distrust).

Since it is well evident that the main source of disagreement cannot be resolved with the data at hand, I will leave it to the other Reviewers and the Editor to decide about the interest of the manuscript to the wide readership of your journal. If published, I will personally be next to certain to cite it – not as hard evidence of the patterns proposed, but as suggestions that they might occur. Whether that suffices for the journal I cannot tell.

Reviewer #4 (Remarks to the Author):

This manuscript adopts a macroecological approach towards addressing how gradients in elevation and land use affect food-web structure. As an additional reviewer, I have been asked to assess the author response to the criticism raised by reviewer 3 (point 1 below) and also have a new independent assessment (further points). Overall, I found the topic of the study intriguing and important. In my opinion, this study presents an important avenue for future research towards marrying macroecology and food-web research. I applaud the authors for directing their research in this direction.

1) The methodology used by the authors establishes a meta-food-web based on expert knowledge

and published links. The local communities are represented by species lists and the links are added from the metaweb. I see the points raised by reviewer 3 that based on this methodology none of the local food webs are accurate as some of the links of the metaweb might not necessarily be realized locally. However, I fully agree with the authors that applying this type of rigor to our science would make most of food-web research impossible. In my opinion, the authors follow a standard procedure and the only way possible to bring food-web research to a macroecological level. Following the arguments raised by reviewer 3 would imply measuring millions of links locally, which is logistically impossible.

Now, the question is whether the results are severely impacted by this inaccuracy. In other studies, I have seen sensitivity analyses that could also be applied here. For instance, the local assemblies of the food-webs could be replicated with e.g. a 10% removal of links (representing links that are locally not realized). Would this lead to similar conclusions?

In addition, I have been going through the appendix and conclude that it includes everything that I would find necessary to replicate the study.

In conclusion, I am strongly leaning in the direction to support the response by the authors related to the methodological criticism raised by reviewer 3.

2) While reading the introduction, I was a bit surprised by the lacking motivation why the gradients of elevation and land use have been chosen. The introduction is not leading to hypotheses and the motivation why the five food-web parameters have been chosen is not evident. I would strongly suggest to either re-write the introduction that it leads to hypotheses related to the food-web parameters chosen or include a larger suite of food-web descriptors (for instance, I am missing anything related to trophic levels which is typically the second important dimension of food-web topology). In the latter case, a fruitful avenue might be to carry out a PCA analysis on the food-web parameters and enter the PCA axes in the subsequent analyses.

3) In the discussion, elevation is related to temperature. I did not find any argument why temperature has not been included as an explanatory variable in the analyses to disentangle its effects from other effects of elevation. This would be a much stronger analysis.

4) Throughout the discussion and conclusion, it is one of the main findings that food-web structure responds to land use but it is not evident what patterns this implies. Is this mainly driven by differences between urban and forest food webs? Can you zoom in and give more specific examples on the dominant effects?

5) A rather minor point but the finding that land-use types affect food-web structure are not novel (e.g. Digel et al. 2014, *Oikos*, <https://doi.org/10.1111/oik.00865>)

I hope that these points help in revising the manuscript. I am quite excited to see food-web research moving in this direction.

Response Letter

Original comments from the Reviewers are in **black** and our responses in **blue**. Corresponding revisions are highlighted with **yellow background** in the manuscript.

Comments from Prof Shawn J. Leroux (Reviewer #1):

The authors have completed a comprehensive revision that addresses all concerns raised by the three reviewers. I appreciated the additional details on methods and justification for data and analysis choices. In addition, the new analyses on connections between green and blue webs and non-linear relationships was well done – thank you.

Response: We thank the Reviewer for this supportive feedback. The Reviewer's constructive comments in the previous round have helped us improve the manuscript a lot. We are happy to learn that the Reviewer is satisfied with our revision.

The authors also provide a useful rebuttal to the concerns of Reviewer #3. I fully agree with the authors on all points. Food webs are a known unknown – we can never know all possible interactions and this is why we have developed many ways to quantify potential interactions. Even if a group was capable of empirically sampling nearly 1000 sites (for many years each!), they would still only capture a subset of all possible interactions. Yet, we are inherently interested in how food webs change across space and time. The authors use a metaweb to define “potential” interactions and they are explicit about the limitations of this approach. To me, this is a robust use of a metaweb and the clear communication of its use is a real strength of the paper – the authors identify the limitations and interpret their findings in light of this. To me, this is more robust than authors considering their empirically intensive sampling to represent “truth”. Most importantly, if a reader disagrees with the metaweb, the authors have provided all the data and references to support the links in their web. Their Rdata files are crystal clear and make this work reproducible. It seems Reviewer #3 may simply be against food web studies or spatial food web studies as they have not provided any concrete suggestions to address the various issues they identify. A probabilistic approach is not the solution here as the authors have clearly argued.

Response: We appreciate the Reviewer's comments supporting our work as regards to the challenge of Reviewer #3 and specifically supporting our reasoning of the metaweb. We totally agree with the Reviewer that the metaweb approach as used here represents the only realistic way to consider potential interaction networks at this large scale across so many species and sites. We are also very thankful for the specific comment on the “crystal clear” codes and data in our Rdata file, which allows reproducibility of our work.

We much appreciate such a firm support.

Comments from Prof Leonardo A. Saravia (Reviewer #2):

I congratulate the authors on doing a thorough job of addressing the points I raised in the previous review round and making a solid defence of some of their methods. I think that the manuscript is generally improved. I would like to repeat that the topic is important and interesting, and will enhance our knowledge about food webs changes over natural and human induced environmental ranges. Thus the ms will be of interest to a broad readership, and it's worth being published in *Nature Communications*.

Response: We thank the Reviewer for this recognition and very supportive comments. We appreciate the methodological input by the Reviewer from the previous round, which greatly helped us make this work conceptually and statistically more concrete.

Comments from Reviewer #3:

The authors have now revised their interesting manuscript, and provided a strong rebuttal of the main criticism advanced. As I am Reviewer #3, I note that the main rebuttal of my own critical comments frames them as being based on a misunderstanding – which I do not think they were.

Our main source of disagreement can be simplified into two key points:

First, I claim that link structure, and thereby food web structure, cannot be inferred from species co-occurrence alone (which is the key assumption behind the “metaweb approach”): two species occurring the same place is simply insufficient proof that they interact. And if we lack such proof, then the actual data is on species pools (which can be conveniently analysed) and not on network structure (as the authors claim and model).

Second, I claim that a metaweb based on directly observed and published evidence of who interacts with whom cannot be replicated if and since it has been mixed with an unknown and undocumented proportion of expert opinion.

In response, the authors frame my criticism as being “a generic critique based on personal opinion, beyond [their] actual work, even though this approach represents an emerging research direction”. I respectfully disagree with them as respectfully as they disagree with me, and – most importantly – see no more trace of a personal opinion in the critique advanced than in the responses proffered.

The first main defence of the current approach is that “other people do the same” – with which I agree. (In fact, I will volunteer a key reference from last week to add to the author’s list of other people who did the same, i.e. Bauer et al. 2022 Ecology Letters). The crux here, though, is that pointing to some precedent does not validate an approach, since other people can make errors, too. Just look at what GBIF data have been used for by happy number crunchers just “because they are there”... What is more, the validity of an approach will vary with what it is used FOR.

The second main defence is that “this is the best we can do since surely, we cannot realistically collect direct evidence on the phenomenon that we claim to be analysing” [which is food web structure]. I accept that this is the case, but if we cannot realistically collect direct observations of something, then that does not in itself justify the use of other data to approximate it.

The third main defence is that the reconstruction of the metaweb can be repeated based on the description provided. I do not see how, since links based on expert opinion cannot be separated from links based on direct evidence? In support of the validity of their data, the authors advance the classical argument of “trust us, we are experts” – which I fully believe, but which is still off the point.

The fourth main defence is that the assumptions are clearly spelled out, and that it is then up to the reader to judge. With this I partly agree: the assumptions are clearly spelled out at the outset of the paper. My claim is still that they are unlikely to be true, and that this undermines the conclusions – which are not framed as being conditional on strong assumptions.

The authors’ claim is then that 1) these are assumptions which simply have to be made for us be able to approach the focal topics with the data at hand [with which I fully agree], and 2) that if they are true then blue and green food webs exhibit all sorts of structural and ecological properties along elevation and among various land-use types (which is very cool if true).

What I believe is that the data suffice for analysing environmental imprints on blue versus green species pools, whereas the authors believe that they suffice for analysing network structure. Since the real data needed to show who is right are critically missing (which is indeed my whole point), I guess the argument boils down to i) whether or not the main assumptions are stated clearly enough, ii)

whether or not the key findings are consistently framed as being conditional on these assumptions, all the way to the concluding section (on which I disagree), iii)) whether or not the key assumption are likely to hold true (which I doubt), and iv) whether or not the results are robust to deviations from the main assumptions (which I distrust).

Since it is well evident that the main source of disagreement cannot be resolved with the data at hand, I will leave it to the other Reviewers and the Editor to decide about the interest of the manuscript to the wide readership of your journal. If published, I will personally be next to certain to cite it – not as hard evidence of the patterns proposed, but as suggestions that they might occur. Whether that suffices for the journal I cannot tell.

Response: We thank the Reviewer for their new comments. When addressing their criticisms, we have improved the way framing our study (e.g., with restructured introduction), which we really appreciate. We respect the different viewpoint, but we believe that we have adopted the best possible approach.

The metaweb is the only method available to study interactions at the scale considered, and we have applied it based on very extensive and solid datasets. Given that three other Reviewers, especially the new Reviewer #4 requested by the Editor to directly compare our responses to the comments of Reviewer #3, were highly supportive of our work, we believe that we have done as much as could be done to justify our work. We followed Reviewer #4's suggestion to perform a sensitivity test to support the robustness of our findings (see below point iv). Beyond that, we feel there is no more analysis we can do to resolve the difference in viewpoints. Thus, as the Editor suggested, we rather revised in the manuscript to our best to further expand on the assumptions and possible limitations of our approach (L98 & L383 onwards).

Here, we would like to still highlight important aspects of our study and respond to the Reviewer's above comments.

i) Firstly, as we have replied in the previous round, and other Reviewers have agreed, the information provided is sufficient making our study reproducible, and all code is given. Thus, we here expand mainly to reply to the Reviewer's concern on whether food-web structure is examinable given the data we complied.

ii) Secondly, while we agree with the Reviewer that “two species occurring the same place is simply insufficient proof that they interact” (quoted), we have to again point out that we did not infer interactions from species co-occurrence alone. In contrast, species interactions were derived from a search of primary observation-based literature, and themselves are data as well.

Our method represents the combination of “species co-occurring” and “reported interacting” data that formed the realistic boundaries of our trophic links thus food-web inferences, not any of them alone. We do agree (and acknowledge throughout the manuscript) that the food webs were inferential/potential ones, but emphasising that they emerged from systematic and coherent comparisons across gradients within the same constraints set by abovementioned realistic boundaries.

We also agree that having empirical measures of every localised trophic interaction would be ideal; but as other Reviewers also acknowledged, this is unrealistic at a macro-ecological level. Extrapolating reported trophic relationships is arguably the most realistic next choice, since these are interactions occurred within a normal (natural) range of species trait-matching or any other lower-level mechanisms that shape species' diets.

We see the Reviewer's point, but we share the same opinion with Reviewer #4 as they commented further below: Applying this type of rigour would make most macro-ecological researches (not only food-web ones) impossible. Every bit of up-scaling inevitably includes simplification and extrapolation

of information. This applies to ubiquitous approaches and topics, such as SDM, remote sensing, and metabarcoding in spatial ecology or BEF research. Our approach is strongly data-driven and thus already at the conservative end of making extrapolation. When we framed the Reviewer's previous comments as "generic critique beyond our actual work", this was not meant as a personal comment, and we were not trying to downplay them—simply because such rigour indeed challenges macro-ecological studies way beyond our piece of work *per se*.

iii) Thirdly, all assumptions and possible limitations of our approach have been clearly acknowledged in the manuscript, and we carefully interpreted our findings accordingly. Among us authors, we have revisited these assumptions and relevant statements, and we are convinced that they hold up, as also backed-up by extensive natural history information on these systems. We do acknowledge that food-web structure could be driven by species composition and other factors causing intra-specific (among-location) dietary variation, and our data could address only the potential outcome of the former (as stated L372 onwards).

iv) Finally, we expanded interpretation of our findings with no deviation from the above limitations. We now, following the suggestion of Reviewer #4, additionally performed a sensitivity test to show that our findings were robust to a potential inaccuracy of the metaweb (added to Section S5).

While the Reviewer #3 disagrees on some aspects of our study, we still would like to thank for their efforts and time of giving rounds of feedback on our work. We acknowledge that we may not reach a consensus on all aspects, yet we appreciate their openness, fairness, and finally also the statement that they will likely refer to our work and the patterns we present. We see this as a constructive and fair scientific process, despite some different views.

Comments from Reviewer #4:

This manuscript adopts a macroecological approach towards addressing how gradients in elevation and land use affect food-web structure. As an additional reviewer, I have been asked to assess the author response to the criticism raised by reviewer 3 (point 1 below) and also have a new independent assessment (further points). Overall, I found the topic of the study intriguing and important. In my opinion, this study presents an important avenue for future research towards marrying macroecology and food-web research. I applaud the authors for directing their research in this direction.

Response: We thank the Reviewer for their positive feedback, particularly the recognition of the importance of our research direction.

We are glad to see that the Reviewer #4 (as well as Reviewers #1 & #2) are strongly in agreement with our view regarding our research approach. We have now again responded and revised the manuscript with respect to Reviewer #3's criticism (also point 1 below). Yet with the Reviewer's support, we feel that this issue is resolved. We also have addressed all their additional points, and revised the manuscript accordingly. We much appreciate their thorough assessment and constructive suggestions that improved the manuscript.

1) The methodology used by the authors establishes a meta-food-web based on expert knowledge and published links. The local communities are represented by species lists and the links are added from the metaweb. I see the points raised by reviewer 3 that based on this methodology none of the local food webs are accurate as some of the links of the metaweb might not necessarily be realized locally. However, I fully agree with the authors that applying this type of rigor to our science would make most of food-web research impossible. In my opinion, the authors follow a standard procedure and the only way possible to bring food-web research to a macroecological level. Following the arguments raised by reviewer 3 would imply measuring millions of links locally, which is logistically impossible.

Response: Thanks for this strong support and agreeing that we were following a standard procedure, and that there is no logistically possible alternative to our chosen approach.

Now, the question is whether the results are severely impacted by this inaccuracy. In other studies, I have seen sensitivity analyses that could also be applied here. For instance, the local assemblies of the food-webs could be replicated with e.g. a 10% removal of links (representing links that are locally not realized). Would this lead to similar conclusions?

Response: This is a good suggestion and input to strengthen the base of our approach and discovery. We have now performed the suggested sensitivity analyses with removal of links, and added them to the manuscript. Specifically, the results show that our original findings were robust to the potential inaccuracy of our metaweb caused by links that are not realised locally (i.e., by removing 10% of the links randomly). Relevant information has been incorporated into our main text (L384 & L527) and SI (Section S5).

In addition, I have been going through the appendix and conclude that it includes everything that I would find necessary to replicate the study.

Response: We much appreciate that the Reviewer (as well as other of the Reviewers) particularly highlighted the completeness and reproducibility of our study with the materials we provide.

In conclusion, I am strongly leaning in the direction to support the response by the authors related to the methodological criticism raised by reviewer 3.

Response: We thank for the Reviewer's strong and firm support for our methodology and arguments made in the previous round. Also, for confirming the completeness of materials that we provided.

2) While reading the introduction, I was a bit surprised by the lacking motivation why the gradients of elevation and land use have been chosen. The introduction is not leading to hypotheses and the motivation why the five food-web parameters have been chosen is not evident. I would strongly suggest to either re-write the introduction that it leads to hypotheses related to the food-web parameters chosen or include a larger suite of food-web descriptors (for instance, I am missing anything related to trophic levels which is typically the second important dimension of food-web topology). In the latter case, a fruitful avenue might be to carry out a PCA analysis on the food-web parameters and enter the PCA axes in the subsequent analyses.

Response: Thanks for this input—this is a fair comment, and we have thought extensively about it, and have addressed it in the manuscript. We fully agree with the Reviewer that the motivation of looking at elevation and land use as the chosen environmental factors should be spelt out and formulated in a more hypothesis-oriented manner.

We have now revised our *Introduction* accordingly (L55, L86, & L112 onwards), to better motivate such gradients selection due to their known influences on ecological communities, and their applicability with the data we have. Yet, we also think that in the field of macro-ecological scale food-web research, there is not yet established theory or reasoning that directly allows us to make more-specific hypotheses about how blue and green food webs would respectively respond to these environmental factors. Simply speaking, such hypotheses would be strongly post-hoc, and we thus mostly discussed them in the *Discussion*. This was also why we were cautious at making more-specific hypotheses in the *Introduction*—our work is pretty much the first exploration of such comparison, and we would like to avoid “inventing” one-to-one predictions or hypotheses based on a retrospect of our results.

Also, thanks for asking about the five food-web metrics chosen. We have now strengthened the reasoning of focusing on them in the *Introduction* and *Methods* (L109 & L475). In brief, these are metrics that concern the holistic structure/function of the network beyond its node- or link-level properties, and we consider the former are more relevant to the processes that may have shaped food webs across a macro-ecological scaled landscape (e.g., community assembly). These metrics were also chosen and analysed similarly in the literature (as cited L478 & L493). We thus consider following the same research design would allow potential cross-study comparison or validation. While node-level properties (e.g., trophic level, as the Reviewer mentioned) are quite meaningful when taking a node-centric perspective, they can easily lose their ecological relevance when being scaled-up to represent the whole network (e.g., by taking the mean or maximum level). They are also intrinsically sensitive to the presence/absence of one or a few other connected nodes, and such sensitivity is not ideal given our local food webs were already products of inference.

3) In the discussion, elevation is related to temperature. I did not find any argument why temperature has not been included as an explanatory variable in the analyses to disentangle its effects from other effects of elevation. This would be a much stronger analysis.

Response: Apologies for this potential unclearness in our manuscript: Temperature was actually included as an explanatory variable in our models and analysed. Elevation and temperature are essentially highly correlated. Yet, when the residual effect of temperature (accounting for elevation) was looked at (i.e., disentangling effects of temperature from elevation), the residual temperature did

not contribute as a good predictor of focal food-web metrics. In other words, by looking at elevation, we already captured most of the temperature influences, meanwhile still included those of other non-temperature elevational drivers. While this was already included in the previous version, we have now made it much more prominent as it is indeed an important point. Relevant content is now revised in the *Methods* (L441, corresponding analyses in SI Table S4) and also when we addressed elevation vs. temperature effects in the *Introduction*, *Results*, and *Discussion* (L112, 140, & L241 onwards).

4) Throughout the discussion and conclusion, it is one of the main findings that food-web structure responds to land use but it is not evident what patterns this implies. Is this mainly driven by differences between urban and forest food webs? Can you zoom in and give more specific examples on the dominant effects?

Response: Thanks for pointing this out, and we happily clarified. We have now restructured some contents of the manuscript to better highlight the key findings regarding the effects of land use:

(i) The different elevational food-web patterns between blue and green food webs were mostly contributed by the webs in farmlands (i.e., in farmlands the blue-green difference is most pronounced; this is now highlighted in L148, L163, & L214).

(ii) There were different elevational patterns between farmland and forest green webs within the same elevational segment (i.e., farmland green webs respond differently to elevation from the forest ones; L307).

These findings suggested land use is influential, and particularly that anthropogenic land use (e.g., agriculture) and climate change (if taking elevation as a climate proxy) can have interactive ecological impacts. We have now revised especially in the *Discussion* to better highlight these found effects (the paragraph L292 onwards).

5) A rather minor point but the finding that land-use types affect food-web structure are not novel (e.g. Digel et al. 2014, *Oikos*, <https://doi.org/10.1111/oik.00865>)

Response: Thanks for this input and we have now incorporated this literature (L294).

I hope that these points help in revising the manuscript. I am quite excited to see food-web research moving in this direction.

Response: The Reviewer's feedback certainly did help and was much appreciated. We are happy to hear about their excitement. We again thanks for their effort.

Reviewer #4 (Remarks to the Author):

The revised version of the manuscript is a pleasure to read. After re-reading my comments to the manuscript, I have felt a bit sorry to come up with additional comments but the new version of the introduction does an excellent job at setting up the questions and methods. Overall, I thus have no further comments other than sending the authors my compliments for accomplishing this excellent study.

Response Letter

Original comments from the Reviewers are in **black** and our responses in **blue**.

Comments from Reviewer #4:

The revised version of the manuscript is a pleasure to read. After re-reading my comments to the manuscript, I have felt a bit sorry to come up with additional comments but the new version of the introduction does an excellent job at setting up the questions and methods. Overall, I thus have no further comments other than sending the authors my compliments for accomplishing this excellent study.

Response: We much thank the Reviewer for their compliments, and for the constructive feedback given in the previous round that helped us to arrive at a better manuscript.